# Ferroptosis spreads to neighboring cells via plasma membrane contacts

Bernhard F. Roeck [1,2], Sara Lotfipour Nasudivar[1,2], Michael R. H. Vorndran[1,2], Lena Schueller[1,2], F. Isil Yapici [2,3,4], Matthias Rübsam [2,4,5], Silvia von Karstedt [2,3,4], Carien M. Niessen [2,4,5] & Ana J. Garcia-Saez [1,2,6] ✉

Ferroptosis is a lytic, iron-dependent form of regulated cell death characterized by excessive lipid peroxidation and associated with necrosis spread in diseased tissues through unknown mechanisms. Using a novel optogenetic system for light-driven ferroptosis induction via degradation of the anti-ferroptotic protein GPX4, we show that lipid peroxidation and ferroptotic death can spread to neighboring cells through their closely adjacent plasma membranes. Ferroptosis propagation is dependent on cell distance and completely abolished by disruption of α-catenin-dependent intercellular contacts or by chelation of extracellular iron. Remarkably, bridging cells with a lipid bilayer or increasing contacts between neighboring cells enhances ferroptosis spread. Reconstitution of iron-dependent spread of lipid peroxidation between pure lipid, contacting liposomes provides evidence for the physicochemical mechanism involved. Our findings support a model in which iron-dependent lipid peroxidation propagates across proximal plasma membranes of neighboring cells, thereby promoting the transmission of ferroptotic cell death with consequences for pathological tissue necrosis spread.

Ferroptosis is an iron-dependent form of regulated cell death that is primarily driven by the accumulation of phospholipid peroxides in cellular membranes[1,2]. However, ferroptosis differs from other regulated cell death pathways in that it is triggered when the cellular redox defense systems exceed their antioxidant buffering capacity[3–5]. In contrast to apoptosis, pyroptosis or necroptosis, ferroptotic cell death lacks a terminal executioner protein[6]. While lipid peroxidation has been identified as a key event during ferroptosis, the molecular mechanisms governing the regulation and execution of ferroptosis remain poorly understood[7,8]. Excessive ferroptosis has been associated with a variety of diseases, including neurodegeneration, renal failure, or stroke[9–13]. In addition, induction of ferroptosis holds potential as an anticancer treatment strategy for refractory cancers. Elucidation of the cellular mechanisms that regulate ferroptosis is therefore of great importance for the development of new strategies to combat human diseases.

Differences in polyunsaturated fatty acid (PUFA) mono- and diacyl phospholipids define the susceptibility of cells to ferroptosis[14,15]. In addition, numerous ferroptosis defense mechanisms have been identified that share the common property of directly or indirectly neutralizing lipid peroxides, including a variety of systems such as GPX4-GSH, FSP1-CoQH, DHODH-CoQH$_2$, MBOAT1/2 and GCH1-BH$_4$[16–18]. Other endogenous metabolites have also been found to serve as radical scavengers that can act as ferroptosis suppressors, including several vitamins (K, E, A), 7-dehydrocholesterol, or tetrahydrobiopterin[19–21]. Among them, the GPX4-GSH system is the most powerful anti-ferroptosis defense system[22]. GPX4 is a member of the glutathione peroxidase (GPX) protein family that converts phospholipid hydroperoxides to phospholipid alcohols. This critical role is underscored by the fact that depletion or inhibition of GPX4 triggers ferroptosis[5,23]. GSH is a tripeptide composed of glycine, glutamate and cysteine. Cysteine is the bottleneck in GSH synthesis and

[1]Institute for Genetics, University of Cologne, Cologne, Germany. [2]CECAD Cluster of Excellence, University of Cologne, Cologne, Germany. [3]Department of Translational Genomics, Center of Integrated Oncology Cologne-Bonn, Medical Faculty, University of Cologne, Cologne, Germany. [4]Center for Molecular Medicine Cologne, Medical Faculty, University Hospital of Cologne, Cologne, Germany. [5]Department Cell Biology of the Skin, Center for Molecular Medicine Cologne, University of Cologne, Cologne, Germany. [6]Max Planck Institute of Biophysics, Frankfurt, Germany. ✉e-mail: ana.garcia@biophys.mpg.de

is mainly imported through the xc⁻ system as cystine, the oxidized form of cysteine[24], which is then reduced to cysteine in the cytosol[25].

To add another layer of complexity, ferroptosis has been proposed to propagate through cell populations in vitro, which has been associated with the spread of necrosis in diseased tissues. Accordingly, inhibition of ferroptosis prevented synchronous necrosis of renal tubules, which was exacerbated by Erastin[26]. In addition, cell death in cultured cells treated with ferroptosis-inducing agents occurred in wave-like patterns with spatiotemporal characteristics distinct from other forms of cell death[27,28]. Platelet activation factor (PAF) and derivatives have been implicated in this process[29]. However, the use of drugs to induce or block ferroptosis, potentially affecting all cells in the population, has limited the study of the spread of ferroptosis across cells. As a result, the phenomenon of ferroptosis propagation remains controversial, with key open questions including its molecular mechanisms and contribution to tissue ferroptosis susceptibility.

Here, we developed a new tool capable of controlled induction of ferroptosis in selected cells by combining GPX4 depletion with the advantages of optogenetics[22]. Using this tool, we demonstrate that single ferroptotic cells are able to induce lipid peroxidation and ferroptosis in untreated neighboring cells, which can then further propagate to their adjoining cells in a distance-dependent manner. We show that ferroptosis propagation can be blocked by chelation of extracellular iron or by genetic inhibition of cell-cell contacts through α-catenin knockout, which can be rescued by reconstitution with exogenous α-catenin. Furthermore, we find that increasing the contact area between adjacent cells or bridging cells through a lipid bilayer promotes ferroptosis propagation. Importantly, we reconstitute the propagation of iron-dependent lipid peroxidation in minimal membrane model systems composed of apposed liposomes made of pure lipids, supporting a physicochemical mechanism that does not require additional cellular components. Based on these findings, we propose that iron-dependent lipid peroxidation and ferroptotic cell death can propagate between neighboring cells through their closely apposed plasma membranes depending on α-catenin. Our findings establish cell death propagation as a feature of ferroptosis and provide new understanding of the molecular mechanisms involved.

## Results

### Light-induced degradation of GPX4 induces ferroptosis with high spatial and temporal resolution

To develop a tool capable of triggering ferroptosis by light-driven depletion of GPX4, we generated an optogenetic construct called Opto-GPX4Deg, in which GPX4 is fused to GFP and LOVpepdegron. LOVpepdegron is a LOV2 domain that changes its conformation upon blue light illumination, exposing an RRRG degron sequence for ubiquitin-mediated proteasomal degradation (Supplementary Fig. 1a)[30,31]. As a negative control, we generated a construct lacking the degron sequence, termed Opto-Ctrl. After confirming their sensitivity to ferroptosis (Supplementary Fig. 1c–h), HEK293 or HeLa cells expressing Opto-GPX4Deg or Opto-Ctrl were illuminated using an optoPlate-96[32,33](see Methods), and the kinetics of cell death in the population were monitored by live cell imaging. HEK293 and HeLa cells expressing Opto-GPX4Deg showed significantly more cell death compared to cells expressing Opto-Ctrl (Fig. 1a, b). The induction of cell death upon Opto-GPX4Deg activation was proportional to the illumination intensity (Supplementary Fig. 1i, j) and could be inhibited by ferrostatin (Fer-1) in a dose-dependent manner (Fig. 1b), supporting ferroptotic cell death.

We validated that activation of the Opto-GPX4Deg tool leads to the degradation of the fusion protein. Western blot (WB) analysis showed that the protein levels of Opto-GPX4Deg, but not the control construct, were significantly reduced (Fig. 1c, d and Supplementary Fig. 2b, c). Interestingly, this trend was also observed for endogenous GPX4 levels, suggesting the existence of coordinated mechanisms for the regulation of GPX4 levels or a dominant-negative interaction of the

degron with endogenous GPX4. As additional evidence, a reduction in GFP intensity in single cells was detected upon illumination of cells expressing the Opto-GPX4Deg, but not in the control construct (Fig. 1e, f, corresponding to experiments in Fig. 1a). Although probably not necessary for its ability to induce ferroptosis, we verified that the Opto-GPX4Deg fusion maintained some level of GPX4 activity by confirming its partial ability to rescue cell death induced by Fer-1 deprivation in GPX4 knockout (KO) cells or by tamoxifen-induced GPX4 depletion (Supplementary Fig. 1l–n). Taken together, these results indicate that the Opto-GPX4Deg tool induces cell death by light-induced degradation of both exogenous and endogenous GPX4.

We confirmed that cell death induced by Opto-GPX4Deg was ferroptotic by lipidomic analysis. Cells expressing Opto-GPX4Deg or Opto-Ctrl were exposed to activating illumination and processed for lipidomics analysis (Supplementary Fig. 2a). WB analysis confirmed the degradation of endogenous GPX4 and Opto-GPX4Deg, but not of the Opto-Ctrl construct (Supplementary Fig. 2b, c). Importantly, the mass spectrometry analysis revealed a significant increase in oxidized lipid species in the Opto-GPX4Deg samples, but not in the negative controls (Fig. 1g). We detected the highest fold enrichments in oxidized phosphatidylcholine (PC) species, consistent with previous work in HeLa cells[34]. These observations were accompanied by a global reduction in fatty acid species (Fig. 1h), in good agreement with lipidomics analysis of ferroptotic HT-1080 cells treated with Erastin, where numerous monounsaturated fatty acids were depleted[35,36].

Combined with live cell microscopy, optogenetic systems allow to study the consequences of activating signaling pathways with high temporal and spatial resolution. As expected, activating illumination of selected, individual HEK293 cells expressing Opto-GPX4Deg, but not Opto-Ctrl, efficiently induced cell death (Fig. 2a, b). We quantified the time to death (using plasma membrane blebbing as a surrogate) of individual cells upon activating illumination, which ranged from 13 to 20 min, with an average of 17 min in Opto-GPX4Deg expressing cells, significantly faster than in negative controls (Fig. 2b). The negative controls also provided an estimate of the photo-toxicity by the activating illumination, confirming that the illumination protocol was not sufficient to induce cell death in Opto-GPX4Deg cells. Treatment with 3 μM Fer-1 rescued cell death to control levels, again supporting that activated cells expressing Opto-GPX4Deg die by ferroptosis. Instead, Fer-1 treatment did not affect the time to death due to photo-toxicity in control samples, nor did it affect cell death induction using an established apoptosis optogenetics tool[37].

To further validate that light-controlled activation of Opto-GPX4Deg induced cell death by ferroptosis, we measured lipid peroxidation in cellular membranes, a hallmark of this form of cell death. We stained cells with C11-Bodipy, a membrane-bound fluorescent indicator that changes emission properties when oxidized and which is commonly used as a proxy to detect ferroptosis-induced lipid peroxidation. Activating illumination caused a wave of C11-Bodipy oxidation in both non-transfected control cells and in cells expressing Opto-GPX4Deg, possibly as a consequence of exposure to high-intensity light. However, the C11-Bodipy oxidation ratio was reduced to normal levels after 1 h in non-transfected cells, whereas the oxidation ratio was maintained or slightly increased in cells expressing Opto-GPX4Deg (Fig. 2c, d). Furthermore, non-transfected cells did not change their morphology over time, whereas cells expressing Opto-GPX4Deg showed typical morphological features of lytic death, such as plasma membrane ballooning (Fig. 2c). Similar results were obtained when measuring lipid peroxidation with C11-Bodipy using the optoPlate-96 (Supplementary Fig. 2d–f). These results suggest that the activating illumination may contribute to lipid oxidation, which would lead to ferroptosis specifically upon Opto-GPX4Deg activation.

Taken together, these data indicate that the induction of cell death by Opto-GPX4Deg is associated with increased levels of lipid peroxidation, providing further evidence for the induction of ferroptosis.

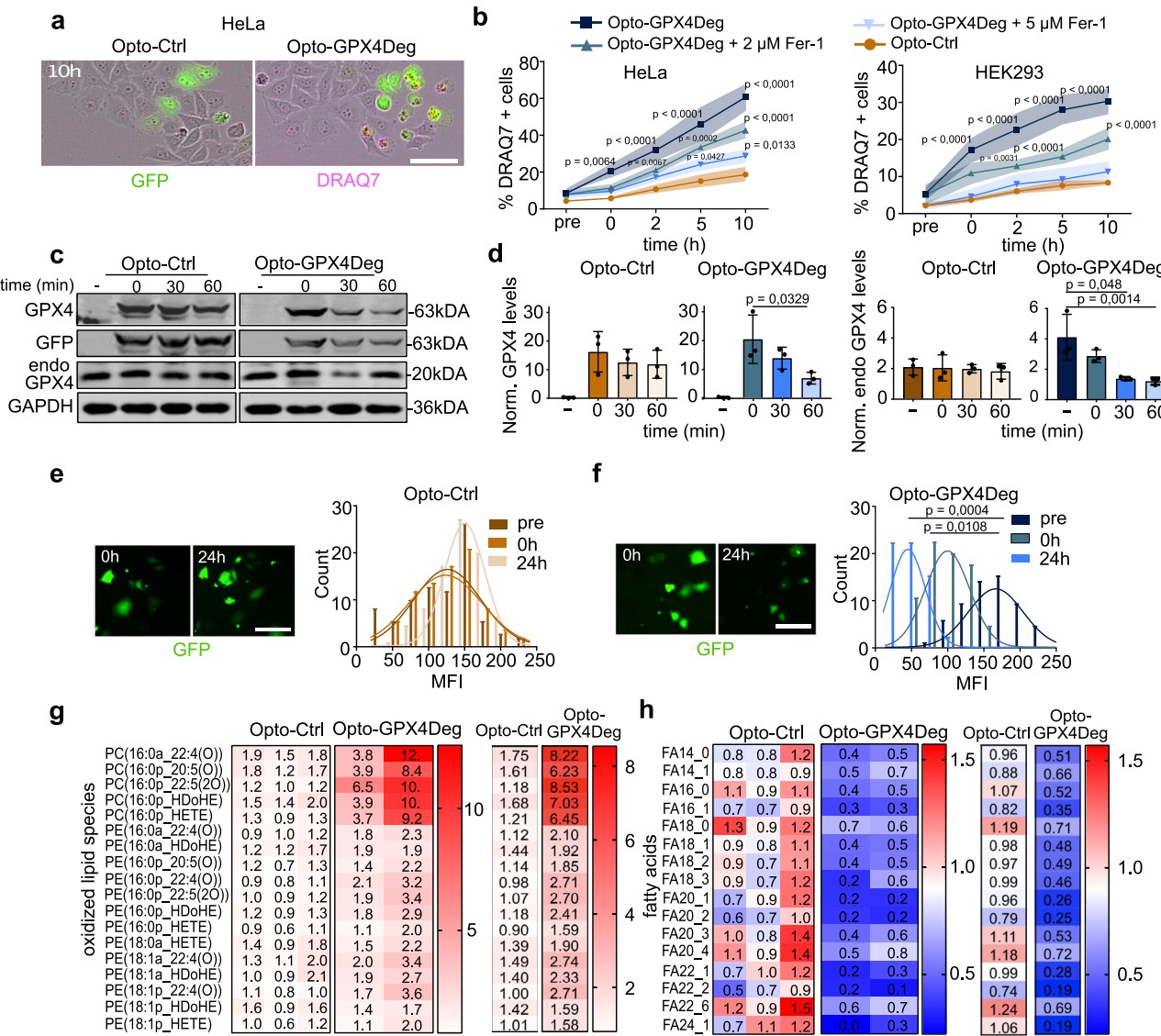

**Fig. 1 | Light-controlled induction of ferroptosis with Opto-GPX4Deg. a** Images of cell death assessment by DRAQ7 staining (magenta) in HeLa cells expressing Opto-GPX4Deg or Opto-Ctrl (green) 10 h post-illumination. Scale bar, 30 μm. **b** Kinetics of cell death triggered by activating illumination in HeLa and HEK293 cells transfected with Opto-GPX4Deg or Opto-Ctrl and treated or not with Fer-1. DRAQ7 positive cells normalized to GFP positive cells. Statistical analysis by two-sided three-way ANOVA corrected for multiple comparisons using Tukey's multiple comparison test. Values displayed as mean ± SD. **c** WB analysis of Opto-GPX4Deg, Opto-Ctrl and endogenous GPX4 protein levels activated or not with indicated illumination times. GAPDH, loading control. **d** Quantification of WBs in (**c**). Protein levels normalized to loading control. Values displayed as mean ± SD. Experiments were performed with three independent biological replicates (*n* = 3).

**e, f** Quantification of the Mean Fluorescence Intensity (GFP) in individual cells at indicated times, as a readout Opto-GPX4Deg degradation (8-bit greyscale). Scale bar, 50 μm. **g, h** Mass spectrometry analysis of oxidized lipids and fatty acids, respectively. Fold change in the heat map calculated by dividing lipid species upon illumination by the non-illuminated control for the individual replicates (left) and average (right). Experiments were performed with three independent biological replicates (*n* = 3) for Opto-Ctrl and two independent biological replicates (*n* = 2) for Opto-GPX4Deg. Each independent biological replicate is represented by the mean of three technical replicates. Unless otherwise stated, statistical analysis by two-sided one-way ANOVA corrected for multiple comparisons using Tukey's multiple comparison test. Exact *p* values are shown. All experiments were performed with three independent biological replicates (*n* = 3) unless otherwise stated.

## Ferroptotic cells are capable of inducing ferroptosis in neighboring cells

Previous studies reported that ferroptosis can spread within cell populations in vitro and contribute to the propagation of necrosis in diseased tissues[26,27]. However, this notion has been difficult to directly demonstrate because conventional drug induction of ferroptosis can affect all cells in the population. We reasoned that the Opto-GPX4Deg system could overcome these limitations and allow the study of ferroptosis propagation. Strikingly, we found that upon illumination of a population of cells transfected with Opto-GPX4Deg, not only cells expressing Opto-GPX4Deg but also non-transfected cells, which we call here bystander cells, died, which did not happen in samples

transfected with Opto-Ctrl (Fig. 3a–d and Supplementary Movie 1). As an additional control to exclude the potential contribution of cells expressing Opto-GPX4Deg below the fluorescence detection range, we activated apparently bystander cells in the Opto-Ctrl and the Opto-GPX4Deg expressing bulk. The percentage of blebbing cells was similar in all conditions, indicating that these cells did not express the optogenetic constructs at sufficient levels to enable light-induced cell death (Supplementary Fig. 1k).

Inspection of the cell death kinetics suggested that, upon activation, cells expressing the Opto-GPX4Deg tool died first, followed by bystander cells (Fig. 3b–d). If the death of bystander cells was related to the activation of Opto-GPX4Deg, one would expect a

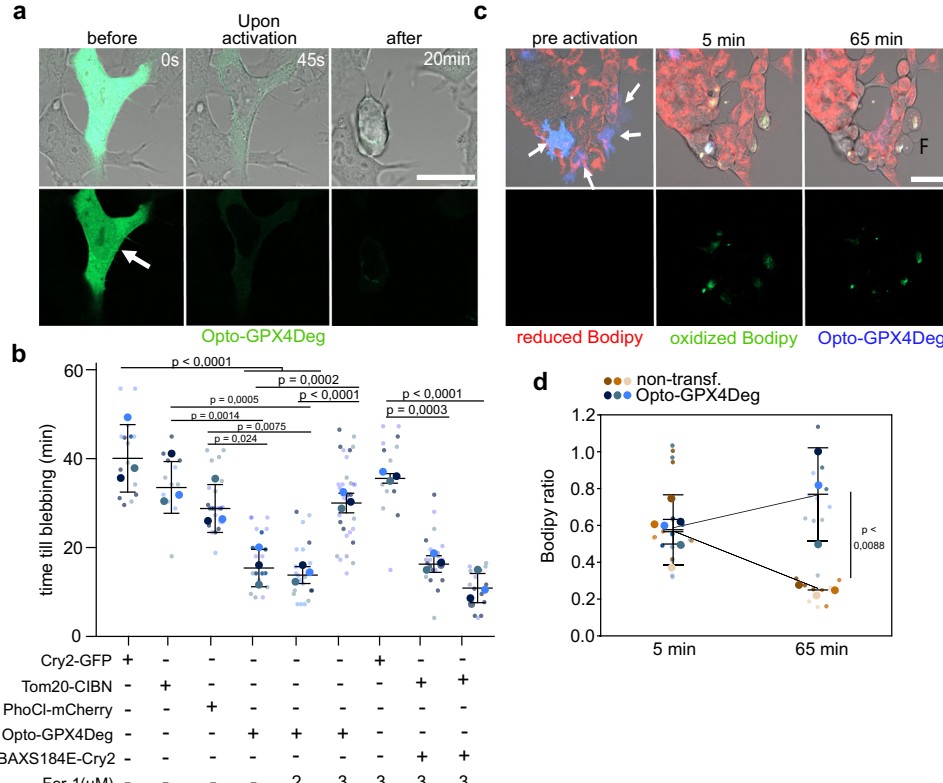

**Fig. 2 | Single cell analysis of ferroptosis induction with Opto-GPX4Deg.**
**a** Images of morphological changes in cells expressing Opto-GPX4Deg upon activation (white arrow). Scale bars, 10 μm. **b** Quantification of the time until blebbing as proxy for cell death for the indicated constructs, treated with Fer-1 as indicated. Big dots represent the replicate averages, small dots represent single cell measurements in individual replicates. Error bars, ±SD. **c, d** Optogenetic activation of Opto-GPX4Deg increases lipid peroxidation assessed by C11-Bodipy oxidation. **(c)** Appearance of green (oxidized) C11-Bodipy fluorescence upon illumination of

selected cells. White arrows, activated Opto-GPX4Deg cells. White asterisk, activated, non-transfected control cells. Scale bar, 50 μm. **d** C11-Bodipy oxidation ratio calculated by normalizing green with green + red C11-Bodipy fluorescence signal. Big dots represent replicate averages, small data dots represent single cell measurements in individual replicates. Values displayed as mean ± SD. Statistical analysis by two-sided one-way ANOVA corrected for multiple comparisons using Tukey's multiple comparison test. Exact $p$ values are shown. All experiments were performed with three independent biological replicates ($n = 3$).

heterogeneous distribution of cell death within the bystander cell population. Accordingly, analysis of the clustering tendency of our data set using Hopkins statistics (see Methods) revealed that cells in the Opto-GPX4Deg samples died preferentially in clusters (value = 0.67), whereas cell death in the population of cells transfected with Opto-Ctrl was randomly distributed (value = 0,43) (Fig. 3e).

To our surprise, we found that the distance from dead bystander cells to dead transfected cells was significantly shorter in samples expressing Opto-GPX4Deg compared to the control (Fig. 3f). This distance was in good agreement with the average diameter of HeLa cells (~23 μm, Supplementary Fig. 2g, h), suggesting that cells dying by ferroptosis induced by Opto-GPX4Deg were able to induce cell death in bystander cells in their immediate vicinity. This behavior was not observed when apoptosis was induced by optogenetics (Fig. 3g), suggesting that the propagation of cell death to neighboring cells is a specific feature of ferroptosis.

We next investigated the mode of cell death experienced by bystander cells adjacent to cells killed by Opto-GPX4Deg activation. We simultaneously quantified the kinetics of membrane oxidation and cell death in transfected and bystander cells, as well as the distances between transfected and bystander dead cells, using an Opto-GPX4Deg construct in which GFP had been exchanged for BFP2. Cells expressing the activated Opto-GPX4Deg tool first accumulated oxidized C11-Bodipy, followed by plasma membrane rupture and cell death. With a delay, cells directly adjacent to these ferroptotic cells also accumulated oxidized lipids and eventually died. Interestingly, this was followed by increased lipid ROS and cell death in their

neighboring cells (Fig. 3h–l, Supplementary Fig. 3 and Supplementary Movie 2). Administration of Fer-1 to samples specifically after Opto-GPX4Deg activation did not block ferroptosis in Opto-GPX4Deg expressing cells, but greatly reduced cell death in bystander cells (Fig. 3k, l; Supplementary Fig. 3a). In control samples, only minor levels of lipid peroxidation were detected and no cell death propagation was observed (Supplementary Fig. 3b and see below). In addition, the distance between dead, bystander cells in the Opto-GPX4 samples, but not in control samples, corresponded to that of neighboring cells (Fig. 3m, n). These results indicate that bystander cells neighboring cells that died by Opto-GPX4Deg activation also underwent ferroptosis, and that they were capable of spreading lipid oxidation and ferroptotic cell death to other bystander cells in their close proximity. This collective evidence thus clearly demonstrates the ability of ferroptosis to propagate across cell populations.

## Disruption of cell-cell contacts abolishes ferroptosis propagation

Since cell death propagation requires cell proximity, we hypothesized that physical contacts between neighboring cells might be directly involved in ferroptosis spread. To disprove this hypothesis, we tested whether conditioned medium containing molecules released from ferroptotic cells would be sufficient to promote ferroptosis in cells. Although we cannot rule out issues of instability or local concentrations, we found that transfer of ferroptotic supernatants onto healthy cells did not result in a death signal in the recipient cells (Fig. 4a).

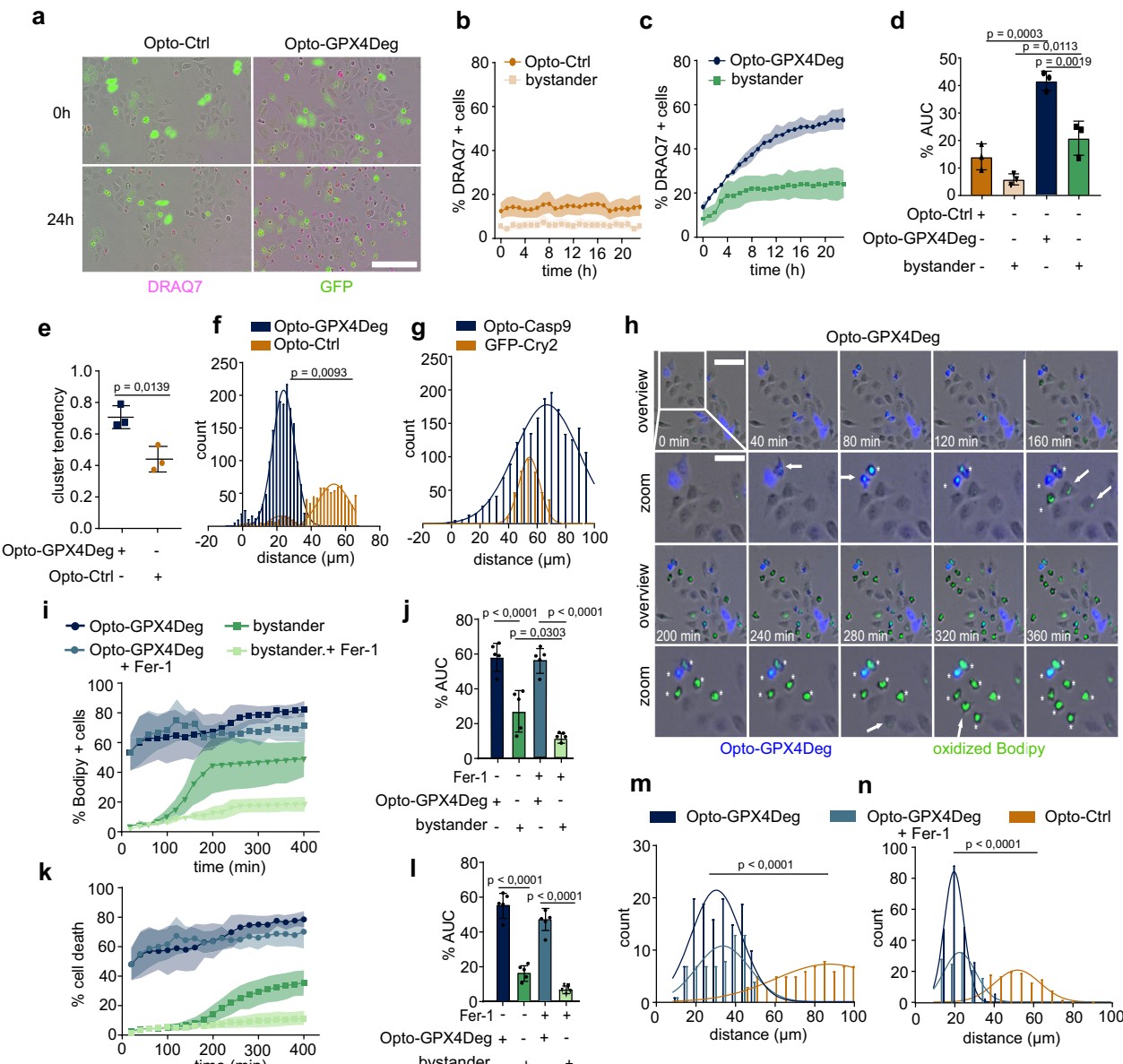

**Fig. 3 | Ferroptotic cells induce ferroptosis in neighboring cells. a** Images of cell death (DRAQ7, magenta) in bystander HeLa cells (GFP negative) at 0 h and 24 h post-illumination in Opto-GPX4Deg and Opto-Ctrl samples. Scale bar, 100 μm. **b**–**d** %DRAQ7 positive cells over time for indicated cell populations from experiments in (**a**). Values displayed as mean ± SD. **d** To express the experimental results in a form that can be more easily statistically compared, we calculated % Area under the Curve (AUC) for the graphs of cell death kinetics of the different cell populations. Two-sided one-way ANOVA corrected for multiple comparisons using Tukey's multiple comparison test. Experiments were performed with three independent biological replicates (*n* = 3). **e** Hopkins statistical analysis for assessment of data cluster tendency. Statistical analysis with a two-sided parametric *t*-test. Experiments were performed with three independent biological replicates (*n* = 3). Values displayed as mean ± SD. **f** Distribution of distances between dead cells expressing Opto-GPX4Deg or Opto-Ctrl and dead, bystander cells from experiments in (**a**–**d**). Statistical analysis by two-sided parametric *t*-test. Values displayed as mean ± SD. **g** Same as in (**f**), but using Opto-Casp9 for light-driven apoptosis induction and respective control construct. Statistical analysis by two-sided parametric *t*-test. Values displayed as mean ± SD. **h** Time series of lipid peroxidation spread. Opto-GPX4Deg, blue; oxidized C11-Bodipy, green. White arrows, cells with C11-Bodipy oxidation. White asterisks, dead cells. Scale bars, 100 μm for overview and 50 μm for zoom. %C11-Bodipy positive (oxidized C11-Bodipy) cells (**i**) and %cell death (**k**) over time in the indicated populations treated or not with 5 μM Fer-1. Values displayed as mean ± SD. **j**, **l** %AUC of the different cell populations in (**h**, **i**). Statistical analysis by two-sided one-way ANOVA corrected for multiple comparisons using Tukey's multiple comparison test. Values are displayed as mean ± SD. Experiments were performed with five independent biological replicates (*n* = 5). Distribution of distances between dead, Opto-GPX4Deg expressing cells and dead, bystander cells (**m**), and distances between dead, bystander cells (**n**). Statistical analysis by a two-sided one-way ANOVA corrected for multiple comparisons using Tukey's multiple comparison test. Exact *p* values are shown. All experiments were performed with three independent biological replicates (*n* = 3) except in (**i**–**l**) and (**f**) five independent biological replicates (*n* = 5) were performed.

Since cytosolic calcium fluxes have been associated with ferroptosis execution[27,38], we next investigated whether calcium could act as a local trigger for ferroptosis propagation to neighboring cells. Consistent with our previous work[38], we observed that Opto-GPX4Deg cells undergoing ferroptosis exhibited increased cytosolic calcium measured with the calcium indicator Fluo-4-AM. However, no significant

calcium influx occurred in neighboring cells during the time window in which they maintained their integrity, suggesting that paracrine calcium fluxes do not play a role in the propagation of ferroptosis (Fig. 4b, c). Interestingly, calcium depletion in the extracellular medium blocked ferroptosis propagation without affecting the intrinsic ferroptosis sensitivity of the cells (Opto-GPX4Deg expressing cells

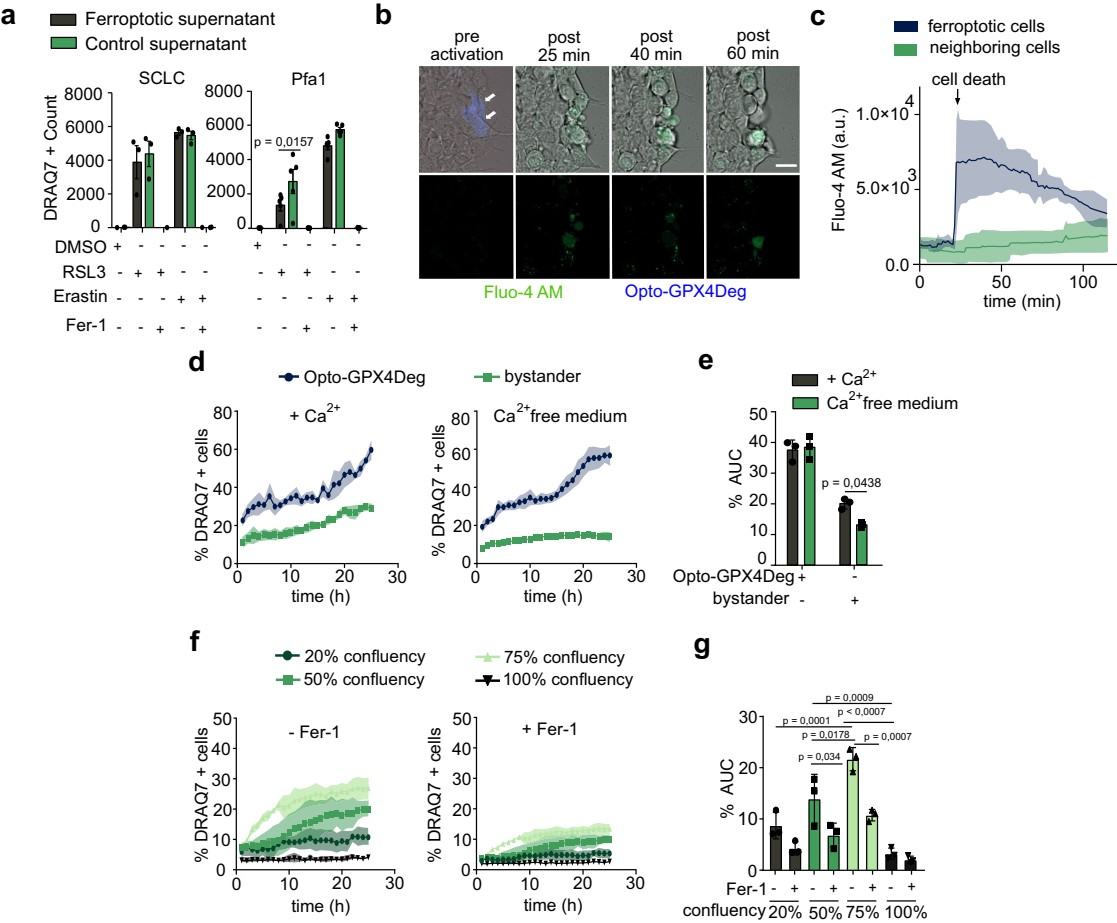

**Fig. 4 | Effect of conditioned media, calcium and cell confluency on ferroptosis.**
**a** Supernatants derived from ferroptotic or control cells from murine small cell lung cancer (SCLC) GPX4 knockout cells under Fer-1 withdrawal (left) or MEFs with inducible GPX4 knockout treated with 1 μM 4-hydroxytamoxifen for 72 h (right) were transferred onto WT MEFs and treated or not with 0.1 μM RSL3 or 1 μM Erastin, and 1 μM Fer-1. Cell death was assessed by DRAQ7 staining and quantified by DRAQ7 object count divided by confluency. Experiments were performed with three independent biological replicates ($n = 3$) except in (**a** right) five independent biological replicates ($n = 5$) were performed. **b** Images of cytosolic Ca²⁺ fluxes in HEK293 cells transfected with Opto-GPX4Deg (white arrows) and stained with 1 μM Fluo4-AM (green) before and after illumination. Scale bar, 50 μm. **c** Single cell quantification of the Fluo4-AM mean fluorescence intensity in (**b**). $n = 5$ ferroptotic

and 40 neighboring cells. a.u., arbitrary units. **d** %DRAQ7 positive HeLa cells over time for indicated cell populations with (left) or without Ca²⁺ (right) in the medium. **e** %AUC for the populations in (**d**). **f** %DRAQ7 positive HeLa cells over time transfected with Opto-GPX4Deg and seeded at the indicated confluency, treated or not with 5 μM Fer-1. Experiments were performed with three independent biological replicates ($n = 3$). **g** %AUC for the populations in (**f**). Two-sided one-way ANOVA corrected for multiple comparisons using Tukey's multiple comparison test. Experiments were performed with three independent biological replicates ($n = 3$). Exact *p*-values are shown. All experiments were performed with three independent biological replicates ($n = 3$) except in (**a** right) five independent biological replicates ($n = 5$) were performed.

died in the same manner). These results indicate that the presence of extracellular calcium plays a role in ferroptosis propagation (Fig. 4d, e and Supplementary Fig. 4a, b), which is interesting because calcium is required to tether extracellular cadherin molecules between cells, suggesting that depletion of extracellular calcium could disrupt cell-cell proximity mediated by cadherins (see below).

Consistent with intercellular distance between cells being a relevant parameter for ferroptosis propagation, we found that increasing cell density rendered bystander HeLa cells more sensitive to cell death induced by neighboring cells dying by Opto-GPX4Deg activation (Fig. 4f, g). As an additional control, RSL3-treated HeLa cells also showed higher sensitivity to ferroptosis at higher cell confluency (Supplementary Fig. 4c, d). However, HeLa cells that reached 100% confluency became insensitive to ferroptosis induction by Opto-GPX4Deg or RSL3, consistent with reports that extremely high cell densities promote the survival of GPX4 (KO) cells[39].

To investigate the role of cell-cell contacts in ferroptosis propagation, we depleted α-catenin, a key component of the cadherin complex that mediates Ca²⁺-dependent cell-cell adhesion[40], in HeLa cells and

performed experiments of light-induced Opto-GPX4Deg activation. Despite partial depletion, the propagation of cell death from Opto-GPX4Deg expressing cells to bystander cells was completely abrogated in the α-catenin knockdown (KD) cells, but not in controls transfected with scrambled siRNA (Fig. 5a–d and Supplementary Fig. 5).

Similar results were reproduced in experiments with α-catenin KD in SCLC cells. However, WT HT-29 cells already exhibited only limited ferroptosis propagation in the presence of α-catenin (Fig. 5e–l Supplementary Fig. 5). Interestingly, we observed that the ability of these cell lines to sustain ferroptosis propagation correlated with E-cadherin expression levels (Supplementary Fig. 6), in line with the protective role ascribed to this protein in ferroptosis[41–43] [https://www.proteinatlas.org/ENSG00000039068-CDH1/cell+line]. Indeed, HeLa cells do not express E-cadherin but present N-cadherin instead, which establishes cadherin-dependent cell-cell contacts also dependent on α-catenin. This apparent contradiction between the anti-ferroptosis function of E-cadherin and its role in establishing cell-cell contacts prompted us to further investigate the mechanism involved.

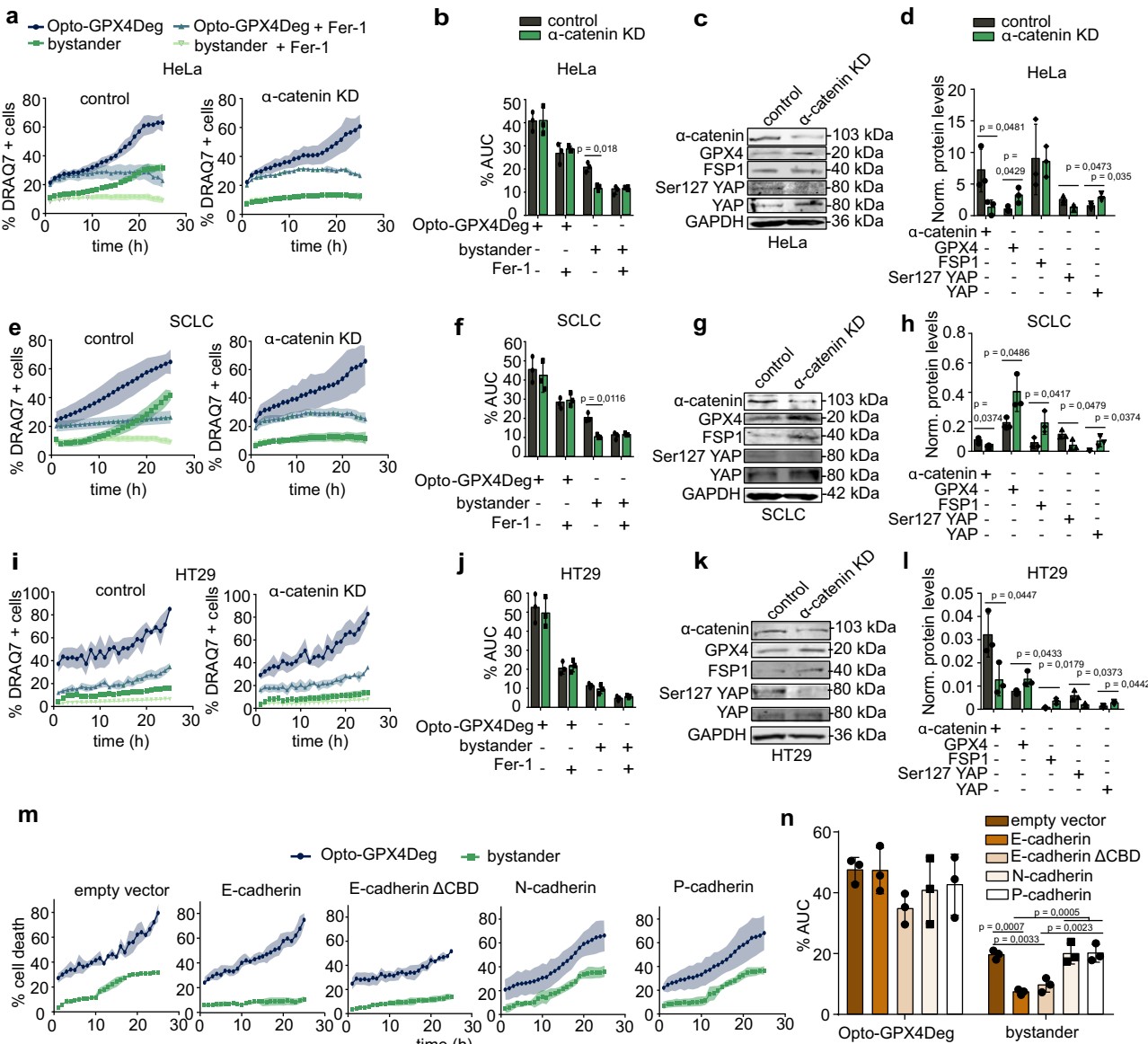

**Fig. 5 | Disruption of cell-cell contacts by α-catenin depletion abolishes ferroptosis propagation.** Opto-GPX4Deg and bystander HeLa (**a–d**), SCLC (**e–h**), or HT29 (**i–l**) cells transfected with either siRNA against α-catenin or scramble siRNA (control), and treated or not with 5 μM Fer-1. **a, e, i** Kinetics of cell death in α-catenin KD and control samples. **b, f, j** %AUC for indicated populations. Experiments were performed with three independent biological replicates (*n* = 3). **c, g, k** WB analysis of indicated protein levels in α-catenin KD and control cells. **d, h, l** Quantification of WB in (**c, g, k**). Protein levels normalized to loading control. Experiments were performed with three independent biological replicates (*n* = 3). **m** Kinetics of cell death in Opto-GPX4Deg and bystander HeLa cells transfected with RFP-tagged empty vector, WT E-cadherin, E-cadherin ΔCBD mutant, WT N-cadherin or WT P-cadherin. **n** %AUC for the cell populations in (**m**) Experiments were performed with three independent biological replicates (*n* = 3). Statistical analysis by two-sided one-way ANOVA corrected for multiple comparisons using Tukey's multiple comparison test or by parametric *t*-test (**d, h, l**). Exact *p* values are shown. All experiments were performed with three independent biological replicates (*n* = 3). Values are displayed as mean ± SD.

In line with the literature[41], transient overexpression of E-cadherin in HeLa cells efficiently blocked ferroptosis propagation without affecting the sensitivity of cells expressing Opto-GPX4Deg to die by light-induced ferroptosis (Fig. 5m, n and Supplementary Fig. 6g–i). In contrast, neither P-cadherin or N-cadherin overexpression prevented the propagation of ferroptosis to bystander cells (Fig. 5m, n and Supplementary Fig. 6g–i). These results suggest a specific role for E-cadherin, but not other cadherins, in blocking ferroptosis spread without interfering with the intrinsic ferroptosis sensitivity of individual cells.

A previous study proposed that E-cadherin depletion promotes ferroptosis by disrupting the cadherin adhesion complex, leading to activation of the transcription factor YAP, which would upregulate pro-ferroptosis factors such as ACSL4 and TFRC[41]. However, this model is not fully consistent with our results. We found that disruption of the

cadherin adhesion complex via α-catenin KD also activated YAP[44], shown by a reduction in phosphorylated YAP and an increase in nuclear YAP (Fig. 5c, g, k and Supplementary Fig. S6), but instead resulted in inhibition of ferroptosis propagation. This was accompanied by an increase in the levels of the ferroptosis protective proteins GPX4 and/or FSP1 (Fig. 5c, d, g, h, k, l). These results indicate that YAP activation induced by disruption of cell-cell contacts cannot explain the anti-ferroptosis role of E-cadherin. Interestingly, we found that overexpression of an E-cadherin mutant lacking the catenin binding domain (CBD) still blocked ferroptosis propagation (Fig. 5m, n). This E-cadherin mutant is unable to efficiently recruit α-catenin to sites of cell-cell contact and thereby does not efficiently link to the actin cytoskeleton, which results in weak cell-cell adhesion. These results reveal that the anti-ferroptotic function of E-Cadherin is

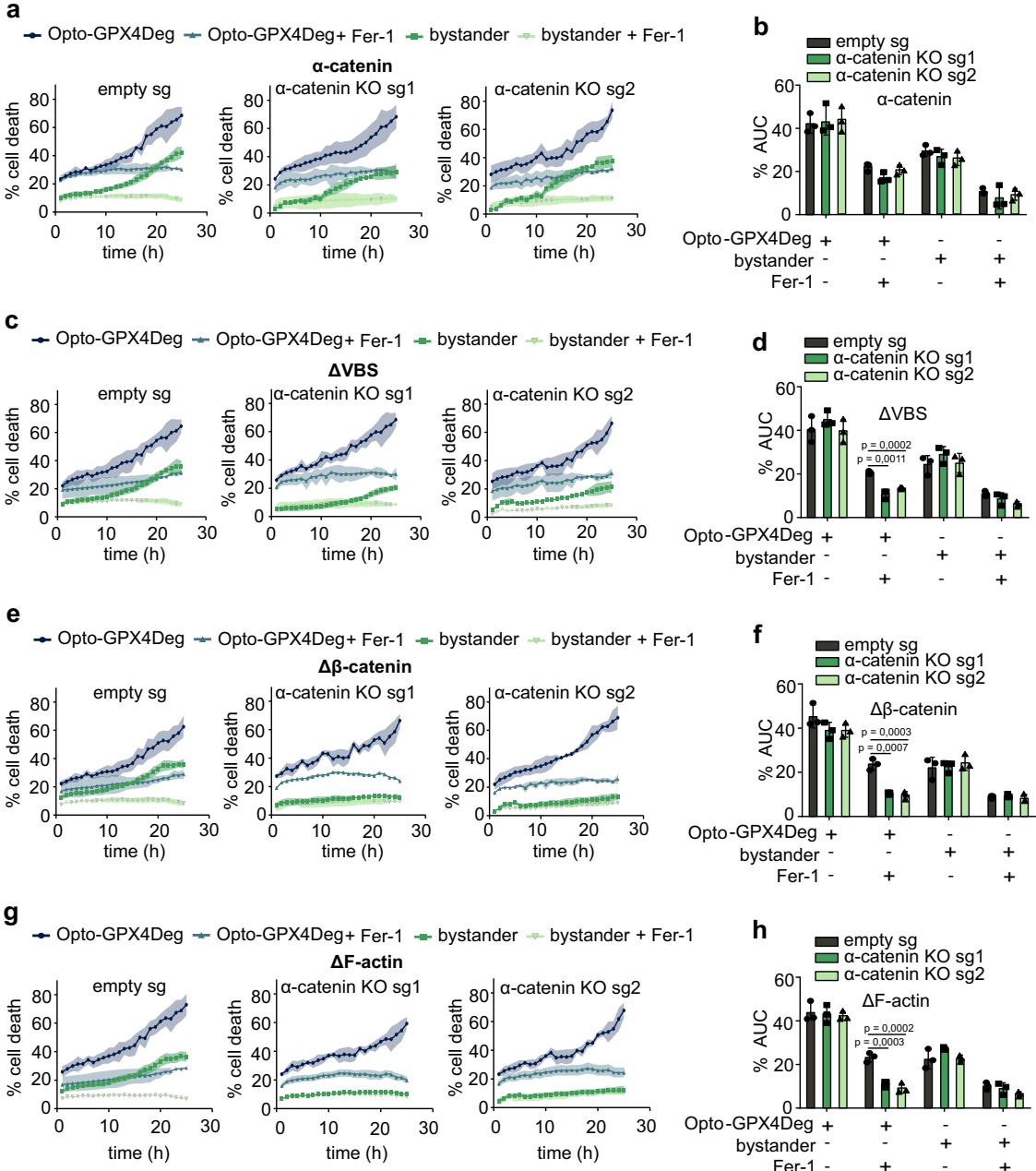

**Fig. 6 | Reconstitution with α-catenin rescues ferroptosis propagation in α-catenin KO HeLa cells. a, c, e, g** Kinetics of cell death for activated Opto-GPX4Deg and bystander HeLa KO clones (sg1, sg2) or CRISPR control (empty sg) reconstituted with exogenous WT or mutant α-catenin treated or not with 5 µM Fer-1. **b, d, f, h** %AUC for indicated cell populations. Experiments were performed with three independent biological replicates (*n* = 3). Statistical analysis by two-sided one-way ANOVA corrected for multiple comparisons using Tukey's multiple comparison test. Exact *p* values are shown. All experiments were performed with three independent biological replicates (*n* = 3). Values are displayed as mean ± SD.

independent of α-catenin recruitment to the complex and likely mediated by other signaling functions of the protein[45,46].

To confirm these findings and to demonstrate that α-catenin is required for the propagation of ferroptosis, we generated α-catenin knockout (KO) HeLa cells and reconstituted them with exogenous WT or mutant α-catenin. We used mutants lacking either the vinculin (ΔVBS), the ß-catenin (Δß-catenin), or the F-actin binding site (ΔF-actin), which partially (ΔVBS) or completely disrupt cadherin-dependent cell-cell contacts. As expected, ferroptosis spreading was abolished in α-catenin KO cells and fully restored when reconstituted with WT α-catenin. Interestingly, reconstitution with the Δß-catenin or ΔF-actin mutants did not rescue ferroptosis propagation, but

reconstitution with the ΔVBS mutant resulted in partial restoration (Fig. 6, Supplementary Fig. 7). Negative controls for toxicity due to illumination under the different conditions are shown in Supplementary fig. 8. In line with this, α-catenin KO cells presented reduced sensitivity to RSL3 treatment, which was enhanced by reconstitution with WT α-catenin and to a lesser extent by reconstitution with the ΔVBS mutant (Supplementary Fig. 9). Although we cannot discard an additional role of α-catenin on ferroptosis that is independent of cell adhesion, since for example α-catenin also modulates the release of extracellular vesicles and membrane budding, these results indicate that genetic interference with the α-catenin function in cell-cell contacts blocks ferroptosis propagation.

## Transfer of iron-dependent lipid peroxidation across apposed membranes mediates ferroptosis propagation

According to our hypothesis that the spread of lipid peroxidation between neighboring cells requires close proximity of their plasma membranes, we reasoned that chemically increasing membrane contact between cells should promote ferroptosis propagation in α-catenin-depleted cells. Accordingly, ferroptosis propagation in Opto-GPX4Deg samples with α-catenin KD was rescued by treatment with non-toxic concentrations of blebbistatin, a drug that increases cell volume by inhibiting cortex contractility, thereby promoting contacts between neighboring cells (Supplementary Fig. 10)[47]. Blebbistatin addition also enhanced ferroptosis propagation in cells treated with scrambled siRNA (Supplementary Fig. 10g–i), as a result of increased cell-cell proximity.

Our hypothesis would also predict that bridging the distance between cells with a membrane capable of sustaining the spread of lipid peroxidation would be sufficient to enhance ferroptosis propagation (Fig. 7a). Indeed, induction of light-controlled ferroptosis with Opto-GPX4Deg in cells seeded on a lipid bilayer accelerated the propagation of lipid peroxidation and ferroptotic cell death compared to cells seeded on glass at the same density, without affecting the ferroptosis sensitivity of individual cells, which could be blocked by Fer-1 addition (Fig. 7b–e, Supplementary Fig. 11a–h and Suplementary Movies 3 and 4). In these experiments, the synthetic lipid bilayers contained unsaturated lipids that could oxidize upon contact with a cell undergoing ferroptosis and then diffuse through the lipid bilayer plane to reach a new cell that had not been induced to undergo ferroptosis, and promote its lipid oxidation and cell death. Accordingly, we could also measure that the lipid bilayer was oxidized by the photoactivation of Opto-GPX4Deg but not control cells (Fig. 7d, e and Supplementary Fig. 11g, h). These results thus demonstrate that the transfer of lipid peroxidation can propagate over distance between cells as long as there is a lipid membrane connecting them.

These results also suggest the existence of iron-dependent lipid oxidation reactions in the extracellular side of the membranes. To test their relevance in ferroptosis propagation, we treated cells with deferoxamine (DFO). DFO is a clinically relevant chelator of extracellular iron (indeed, the synthesis of cell-permeable derivatives of DFO has been the subject of considerable effort[48–51]). Interestingly, we found that DFO treatment inhibited ferroptosis propagation without affecting the intrinsic sensitivity of individual cells to ferroptosis (Fig. 7f–i and Supplementary Fig. 11i–p). In contrast, chelation of both extracellular and intracellular iron with 2,2-dipyridyl blocked ferroptosis in both bystander cells and cells expressing Opto-GPX4Deg. DFO could accumulate over time at lysosomes via endosomal uptake, which might interfere with ferroptosis. Yet, under our experimental conditions, 4 h or 8 h pretreatment with DFO did not alter the cell death levels of activated Opto-GPX4Deg expressing cells compared to the DMSO control (Supplementary Fig. 11k–p). These results would suggest that iron-dependent lipid peroxidation reactions at the outer leaflet of the plasma membrane are required for the propagation of ferroptosis.

Finally, to provide definitive evidence that lipid peroxidation can spread between apposed plasma membranes, we used a chemically controlled minimal system based on liposomes made of pure lipid membranes and devoid of cellular components. We prepared two populations of Giant Unilamellar Vesicles (GUVs), which are micrometer-sized liposomes that can be visualized with confocal microscopy, all containing C11-Bodipy to assess their oxidation state. We included a photosensitizer probe, DMMB, in a population of the liposomes that we called "donor" GUVs, which were also labeled with DiD to distinguish them in the microscopy images. DMMB is a photosensitive lipid that has been shown to induce membrane oxidation in GUVs after exposure to activating illumination[52,53]. The second population of GUVs lacked DMMB and DiD, which we called "acceptor"

GUVs. We brought the membranes of donor and acceptor GUVs into contact by including biotinylated lipids in their composition and streptavidin in the external medium (Fig. 7j and Supplementary Fig. 12). We confirmed that, before illumination, all GUVs membranes were in a reduced state based on the C11-Bodipy fluorescence ratio. We then illuminated a small membrane region of the donor GUV and quantified the appearance of signal from the oxidized form of C11-Bodipy in the donor GUV membrane over time (Fig. 7k and Supplementary Fig. 12a, b). Importantly, at a later stage we were also able to detect an increase in oxidized C11-Bodipy fluorescence in the attached acceptor GUV that depended on the presence of iron in the external medium, indicating iron-dependent propagation of lipid oxidation to the acceptor GUV (Fig. 7l). As controls, GUV imaging without photoactivation did not result in lipid oxidation measured by C11-Bodipy fluorescence (Supplementary Fig. 13). Also, the lack of transfer of DiD form the donor to the acceptor shows that despite contact, their membranes do not exchange lipids and retain their identity (Fig. 7j). Finally, light-induced lipid oxidation in donor GUVs did not propagate to acceptor GUVs if their membranes were not in close contact (Supplementary Fig. S12c). These results demonstrate that close proximity is sufficient for the spread of iron-dependent lipid peroxidation between membranes, and that no other cellular components or pathways are required.

Collectively, these findings reveal a mechanism for ferroptosis propagation across cell populations based on the physicochemical transfer of iron-dependent lipid peroxidation reactions between adjacent membranes.

## Discussion

Here we developed a novel optogenetic tool for the light-controlled induction of ferroptosis. By selectively inducing ferroptosis in a few cells of choice, we could directly visualize the increase in lipid oxidation followed by cell death not only in the activated cells, but also the spread over time to bystander cells that had neither been exposed to a ferroptosis-inducing signal nor genetically modified in any way that could alter their ferroptosis sensitivity. Remarkably, once the cells became ferroptotic, the bystander cells were also able to further propagate lipid oxidation and subsequent cell death to their neighbors. Thus, our results clearly demonstrate that ferroptosis can propagate through cellular populations, as proposed in previous studies[27,54].

Importantly, Opto-GPX4Deg allowed us to dissect the mechanisms underlying the spread of ferroptosis. We found that the propagation of ferroptosis induced by Opto-GPX4Deg was dependent on the distance between cells. In contrast to optogenetically triggered apoptosis, cell death induction was highly prevalent in adjacent cells, identifying proximity as a key parameter in ferroptosis spread. Importantly, ferroptosis propagation was abolished upon α-catenin depletion, thus strongly suggesting a role for the cadherin adhesion complex and cell-cell contacts in this process. The specificity of the role of α-catenin in ferroptosis propagation was confirmed by reconstitution experiments in which WT α-catenin, but not mutants that disrupt efficient cell-cell adhesion, fully restored ferroptosis propagation in bystander α-catenin KO HeLa cells. Furthermore, our data point to a role for cell surface iron-mediated lipid peroxidation in ferroptosis propagation as DFO, a mostly extracellular iron chelator, abrogated ferroptosis propagation without affecting the intrinsic ferroptosis sensitivity of individual cells.

It is worth noting that E-cadherin, an adhesion molecule that drives cell-cell contacts, has a protective role in ferroptosis, which has been proposed to depend on the Hippo pathway and YAP signaling[41–43] [https://www.proteinatlas.org/ENSG00000039068-CDH1/cell+line]. In agreement with this function, we found that ferroptosis propagation was inefficient in HT-29 cells, which have high levels of E-cadherin expression, and could be blocked by exogenous expression of E-cadherin but not P- or N-cadherin in HeLa cells. However, our

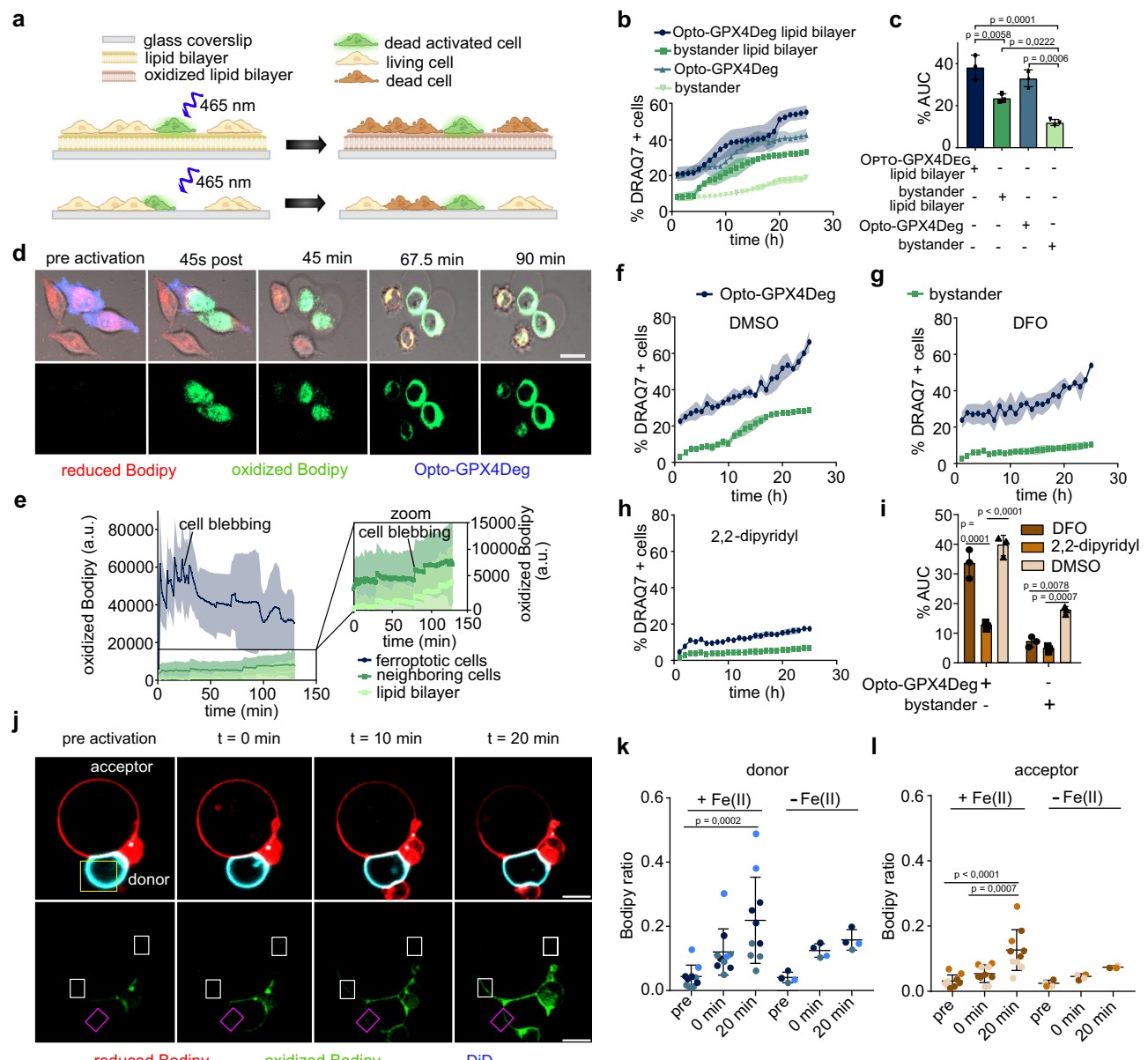

**Fig. 7 | Ferroptosis propagates via transfer of lipid peroxidation reactions between proximal plasma membranes of neighboring cells. a** Scheme of experimental design in (**b**). Opto-Ctrl or Opto-GPX4Deg and bystander HeLa cells seeded on a supported lipid bilayer or directly glass were exposed to activating illumination. The lipid bilayer is meant to act as a membrane bridge between cells and to facilitate the diffusion of oxidized lipids. Created in BioRender. Garcia, A. (2025) https://BioRender.com/v39k921. **b** Kinetics of cell death for activated Opto-GPX4Deg and bystander HeLa cells grown or not on a lipid bilayer and treated or not with 5 µM Fer−1. **c** %AUC for the indicated cell populations. Experiments were performed with three independent biological replicates ($n = 3$). **d** Images of C11-Bodipy oxidation and cell death in activated Opto-GPX4Deg HeLa cells and bystander neighboring cells grown on a lipid bilayer. Opto-GPX4Deg, blue; oxidized Bodipy, green; reduced Bodipy, red. Scale bar, 30 µm. **e** Quantification of oxidized C11-Bodipy over time in cells and in the lipid bilayer (**d**). **f**–**h** Kinetics of cell death for activated Opto-GPX4Deg and bystander HeLa cells treated with DMSO, 100 µM DFO or 100 µM 2,2dipyridyl as indicated. **i** %AUC for the indicated cell populations. Experiments were performed with three independent biological

replicates ($n = 3$). **j** Images of C11-Bodipy oxidation (green) in a "donor" GUV stained with DiD (cyan) and containing a photosensitizer lipid (DMMB) before and after activating illumination. Lipid peroxidation (measured as increase in oxidized C11-Bodipy fluorescence, green) is spread to an attached "acceptor" GUV lacking DMMB. Reduced C11-Bodipy shown in red. Controlled illumination in a small membrane region of the donor GUV (yellow box) induced light-activated lipid oxidation in the donor GUV and later on in the acceptor GUV. $t = 0$ min indicates first image after activation. Membrane regions in which the C11-Bodipy oxidation ratio was calculated are shown in white (acceptor GUV) and pink (donor GUV) boxes. Scale bar, 5 µm. **k, l** Quantification of the ratio between oxidized and reduced C11-Bodipy in donor and acceptor GUVs over time in absence or presence of 10 mM iron (II) perchlorate. Experiments were performed with three independent biological replicates ($n = 3$). Statistical analysis by two-sided one-way ANOVA corrected for multiple comparisons using Tukey's multiple comparison test. Exact $p$ values are shown. All experiments were performed with three independent biological replicates ($n = 3$). Values are displayed as mean ± SD.

combined results indicate that inhibition of YAP activity alone cannot explain the protective role of E-cadherin in ferroptosis. First, both E-cadherin overexpression and α-catenin depletion resulted in YAP activation, as previously shown[44], but had opposite effects on

ferroptosis. In addition, expression of a mutant version of E-cadherin, which is unable to bind α-catenin and can only mediate weak cell-cell contacts, but retains other signaling functions of the molecule, still inhibited ferroptosis while also activating YAP signaling. Given that

E-cadherin is upregulated by increasing cell density[41], it is tempting to speculate that the anti-ferroptotic role of E-cadherin has evolved to counterbalance the increased sensitivity to ferroptosis propagation in tissues with very high cell-cell contacts.

Previous work by us and others showed that ferroptosis induction leads to lipid peroxidation accompanied by a sustained increase of cytosolic $Ca^{2+}$ levels prior to cell bursting[27,38]. Our previous findings suggested that calcium fluxes were caused by membrane damage in the form of small pores at the plasma membrane downstream of lipid peroxidation[38]. Consistent with this, we were able to recapitulate these findings with our optogenetic system. However, the results do not support a role for paracrine calcium fluxes in ferroptosis propagation. While we were able to observe increased cytosolic calcium in cells dying by Opto-GPX4Deg activation, we did not detect significant changes in cytosolic calcium in neighboring cells during the time window in which cell death occurred in the opto-activated cells and once the calcium content of this cell was diluted in the medium after bursting. Instead, we identified a role for extracellular calcium in the propagation of ferroptosis consistent with its role in of cell-cell contacts, since cadherins are calcium-dependent cell adhesion molecules, whose extracellular domain unfolds in the absence of extracellular calcium and becomes sensitive to protease cleavage[55].

Although there may be additional mechanisms based on soluble factors contributing to the propagation on ferroptosis, our collective evidence shows that ferroptosis can propagate to neighboring cells via the transfer of iron-dependent lipid peroxidation reactions through the intimate apposition of plasma membranes enabled by cell-cell contacts. In addition to genetic manipulation of cell adhesion, bringing cell membranes into close contact with blebbistatin is sufficient to rescue ferroptosis propagation in α-catenin-depleted cells. Moreover, bridging the distance between cells with an artificial lipid bilayer that connects their membranes and can spread lipid peroxidation was sufficient to significantly enhance ferroptosis propagation. Finally, using minimal reconstituted systems based on purely synthetic membrane models, we show that lipid peroxidation can propagate between contacting lipid membranes depending on the presence of iron. These experiments directly demonstrate that lipid peroxidation can spread between apposed membranes in absence of any other cellular components, thereby strengthening the evidence for the physicochemical mechanism of ferroptosis propagation uncovered here.

A recent study described that ferroptosis can propagate through ROS trigger waves, which were initiated using blue laser light combined with sublethal cystine uptake suppression across entire cell populations, enabling cell contact-independent propagation[56]. However, under such conditions, cells were artificially primed for ferroptosis and stressed close to the cell death threshold. In contrast, our optogenetic approach selectively induces ferroptosis in a subset of cells without altering the ferroptosis sensitivity of bystander cells. Under our experimental conditions, we did not observe cell contact-independent ferroptosis propagation. Our results indicate that direct cell-cell contacts and iron-dependent lipid peroxidation transfer likely play a predominant role in ferroptosis propagation under more physiological conditions lacking ferroptotic priming.

Ferroptosis spread has been proposed to play a role in the formation of necrotic tissue in intestinal epithelium[57], heart tissue[58–60], excitotoxicity in the brain[61] and renal tubules[26], highlighting its pathophysiological relevance[7,26,62]. Remarkably, our findings demonstrate that ferroptosis propagation can be blocked both genetically and with drugs, and thus can be targeted for therapy. Treatment with DFO, a clinically approved drug for the treatment of iron overload[63] that we show here specifically blocks ferroptosis propagation, has been shown to mitigate ischemic heart and brain injury in rodents[60,64–67]. In clinical studies, DFO ameliorated oxidative stress without reducing heart infarct size[68], protected the myocardium against reperfusion injury during coronary artery bypass[69] and slowed the progression of Alzheimer's disease-associated dementia[70]. The protective role of DFO in these settings also highlights the specific contribution of ferroptosis propagation to disease. Thus, by shedding new light on the molecular mechanism of ferroptosis propagation, our results may guide new therapeutic strategies against ferroptosis-associated diseases. Furthermore, the Opto-GPX4Deg tool presented here provides new opportunities to address the physiological relevance of ferroptosis propagation in different physiological and pathological settings in vivo and to deepen our understanding of how ferroptosis modulators can be exploited for future treatments.

In conclusion, using a novel optogenetic tool for light-controlled induction of ferroptosis, we demonstrate that lipid peroxidation and ferroptotic cell death can propagate to neighboring cells in close contact. We also show that ferroptosis propagation can occur through the transfer of iron-dependent lipid peroxidation reactions between the closely apposed plasma membranes at cell-cell contacts. The ability to abolish ferroptosis propagation genetically or with extracellular iron chelators opens new avenues for specifically targeting ferroptosis spread in therapy.

## Methods
Correspondence and materials should be requested to the corresponding author.

### Reagents
Erastin-1, RSL3, Nec-2 and Fer-1, and were purchased from Biomol (Germany). zVAD was obtained from APEXBIO (Houston, TX, USA) and C11-BODIPY 581/591 as well as Fluo-4-AM from Thermo Fisher. FINO2 was purchased from MedChemExpress (USA) Tamoxifen was bought form Sigma-Aldrich. DRAQ7 was obtained from Invitrogen. Blebbistatin was purchased from Cayman Chemicals. Both DFO and 2,2-Bipyridyl were obtained from Sigma-Aldrich. Lipid standards or dyes were purchased from Sigma Aldrich (St. Louis, US) or Avanti Polar Lipids (Alabaster, US). Organic solvents were purchased from Supelco/Merck KGaA (Darmstadt, Germany).

### Plasmids
Plasmids were cloned via restriction digest cloning as follows: The pMTG02-Amp-FRT plasmid (Addgene #128271) was used as template for amplifying the LOVpep (without the RRRG degron) and LOVpepdegron fragments, which was cloned into the pcDNA™3.1 (+) backbone from Invitrogen using the ApaI and XhoI restrictions sites (primers: Lovpep_xhoI_fwd sequence = tggtatcCTCGAGttttggctactacacttgaac-gtattgag; LOVpep_apaI_rev sequence = atattaaGGGCCCttagccgcggcggcgggg; LOVpepwo_apaI_rev sequence = tattaTAAGGGCCCggcggcctcgtcgatgttctc). Subsequently, the cDNA of GPX4 was amplified from the GPX4 plasmid (Addgene #38797) and inserted in the before generated pcDNA3.1_LOVpepdegron and pcDNA3.1_LOVpep plasmids using the EcoRI and NotI restriction sites. The cDNA of the EGFP was amplified from the pGFP-Cytochrome C (Addgene #41181) and inserted in the pcDNA3.1_GPX4_LOVpepdegron, pcDNA3.1_GPX4_LOVpep and the pcDNA3.1_LOVpepdegron to generate in the paper used GFP_GPX4_LOVpepdegron (Opto-GPX4Deg), GFP_GPX4_LOVpep and GFP_LOVpepdegron plasmids (Opto-Ctrl) (primers: EGFP_HindIII_fwd sequence = atattacAAGCTTCAC CAT-GGTGAGCAAGGGCGAGGAGC; EGFP_EcoRI_rev sequence = atattaca GAAT-TCGACTTGTACAGCTCGTCCATGCCGA). To generate the BFP_GPX4_LOVpepdegron plasmid the mTagBFP2-TOMM20-N-10 (Plasmid #55328) was used as a template to amplify BFP2 and inserted into the GFP_GPX4_LOVpepdegron using the HindIII and EcoRI restriction sites (primers: mTagBFP2_HindIII_fwd sequence = atattaAAGCTTcaccatggtgt ctaagggcgaa; mTagBFP2_EcoRI_rev sequence = atattaG-AATTCgcattaag tttgtgcccccagtttgcta. The GFP_Cry2 plasmid was generated by amplifying Cry2 from the Cry2(1-531)-mCh-BAXS184E plasmid (see below), which was then inserted into the GFP_LOVpepdegron using the NotI and ApaI restriction sites (primers: Cry2_NotI_fwd sequence = attaGCGGCCGCat

gaaga-tggacaaaaagactatagt; Cry2_ApaI_rev sequence = tattaTAAGGG CCCtac-ttgttggtcattagaagaca). The following plasmids were obtained from Addgene: Cry2(1-531)-mCh-BAXS184E (#117238), Tom20-CIB-GFP (#117242), pcDNA-NLS-PhoCl-mCherry (#87691), pmcherry α-catenin WT (#178646) and pmcherry α-catenin ΔVBS (#178649). ΔF-actin was generated by amplifying WT α-catenin 1-864 and Δß-catenin by amplifying WT α-catenin 259-906 subsequently inserting the amplifications into pmcherry α-catenin WT using SacI and MfeI restriction sites (primers: deltaf_SacI_fwd sequence = atattaGAGCTCatgactgctgtccatgcaggc; deltaf_MfeI_rev sequence = atattaCAATTGTTAaccctgtgactttggtatttggtagaggc; deltab_SacI_fwd sequence = atattaGAGCTCATGttgaggaatgctggcaat; deltab_MfeI_rev sequence = atatta-CAATTGttagatgctgtccatagctttg). pcDNA3-mRFP (Plasmid #13032) was used as a control and the pCS2+mEcadRFP and pCS2+mEcad ΔCBD-RFP was a kind gift by C. Niessen, Univ. of Cologne. N-cadherin in pCCL-c-MNDU3c-PGK-EGFP (Plasmid #38153) and pcDNA3 P-cad (Plasmid #47502) were used to amplify N- and P-cadherin in pCS2+mEcadRFP using HindIII/XbaI restriction sites (primers: N-cadherin_HindIII_fwd sequence = atattaAAG CTTcaccatgagggacaattggagaagtg; N-cadherin_XbaI_rev sequence = tatta TCTAGAGATCCTcgataccgtcgagatccg; P-cadherin_HindIII_fwd sequence = tattaAAGCTTatggggctccctcgtggacctctc; P-cadherin_XbaI_rev sequence = tattaTCTAGAGGATCCGtcgctcctccccgccaccg). The insert sequences were validated by Sanger sequencing.

## Cell culture

All cell lines were cultured at 37 °C in a humidified atmosphere containing 5% $CO_2$. HEK293 and HT-29 cells (kindly provided by F. Essmann, Univ. of Tübingen), HeLa (kindly provided by A. Villunger, Med. Univ. of Innsbruck) cells and the tamoxifen inducible GPX4 KO Pfa1 cells (kindly provided by M. Conrad, Helmolz Center Munich) were cultured in low-glucose Dulbecco's modified Eagle's medium (DMEM) (Sigma-Aldrich), and supplemented with 10% fetal bovine serum (FBS), 1% penicillin−streptomycin (P/S) (Thermo Fisher Scientific). RP285.5 murine GPX4 KO SCLC cells (kindly provided by S. von Karstedt, Univ. of Cologne) were cultured in RPMI-1640 (Sigma-Aldrich), and supplemented with 10% fetal bovine serum (FBS), 1% penicillin−streptomycin (P/S) (Thermo Fisher Scientific).

## Cell transfection

For transfections, Polyethylenimine (PEI; Polyscience, Warrington, PA) with a concentration of 1 mg/mL was used in a 5:1 ratio (PEI: DNA) for the confocal experiments and in a 3:1 for the high-throughput optogenetic experiments. In the following, the transfection procedure is described for the 8-well chambers: 25 μl Opti-MEM (Thermo Fisher) per well was mixed with 1.2 μl PEI and incubated for 2 min at room temperature. 240 ng plasmid was mixed with 25 μl Opti-MEM and subsequently added to the PEI/Opti-MEM mixture. Then the mixture was resuspended and incubated for 20 min at room temperature. Then the media from the 8-well chambers was aspirated and replaced with 200 μl DMEM media without phenol red, supplemented with 10% fetal bovine serum (FBS), 1% penicillin−streptomycin (P/S) (Thermo Fisher). Afterwards the transfection mix was added dropwise to the wells. For transfecting cells e.g., in the 6-well plate or 96 well-plate format, the above mentioned volumes of reagents were proportionally adapted and in those formats the cell culture media was not replaced on the next day, but after the transfected fresh media was supplemented with respective drugs or/and cell death markers and added to the wells.

## Silencing experiments

siRNA transfection was performed in six-well plates. For every well, 20 pmol of α-catenin siRNA (GGAGCCAGCUAGAUAUUAA, SiTools Biotech) or siRNA control/non-targeting (D-001810-0120, SiTools Biotech) was premixed in 100 μl Opti-MEM (Thermo Fisher) and 1:1 mixed with 5-μl Lipofectamine RNAiMAX (Invitrogen) or PEI (see cell transfections) diluted in 100 μl Opti-MEM. The mixture was incubated

for 25 min at room temperature and afterwards the mixture was added dropwise to the cells. siRNA transfection was performed for 48 h prior to the transfection with the optogenetic tools or for 72 h before harvesting cells for WB. The levels of the endogenous protein were checked by immunoblotting.

## CRISPR/Cas9 cell line generation

α-Catenin CRISPR/Cas9 knockout were generated in HeLa WT cells. For CRISPR transfection $7 \times 10^5$ cells were seeded in a six-well plate, where after a reverse transfection was performed with 1,5 μg CRISPR construct and 6 μl PEI. Sixteen hours after transfection single CRISPR expressing cells (GFP positive) were sorted into 96 well plates and cultured for validation. Knockout success was validated using western blotting.

Plasmid and constructs: the following guide RNA sequences were used for the generation HeLa α-Catenin CRISPR/Cas9 knockout cell lines:

hCTNNA1_sgRNA#1FWD CACCGTTTATCGATGCTTCCCGCC hCTNNA1_sgRNA#1REV AAACGGCGGGAAGCATCGATAAAC

hCTNNA1_sgRNA#2FWD CACCGTCACGTAGTCACCTCAGAGA hCTNNA1_sgRNA#2REV AAACTCTCTGAGGTGACTACGTGAC

Pairs of oligonucleotides containing the gRNA sequence were cloned into the pU6-(BbsI)sgRNA_CAG-Cas9-venus-bpA (Addgene #86986) using the restriction site BbsI.

## Confocal live cell imaging

According to workflow in Fig. S1b, 5.000 cells were seeded into removable 8-well chambers (Cat.No: 80841 ibidi) in combination with a glass cover slip suitable for confocal microscopy (VWR). All confocal experiments were performed using a Confocal Laser Scanning Microscope LSM 980 with Airyscan 2 and multiplex, GaAsP (4x), PMT (2x), T-PMT, external BiG.2 Typ B 980 detector (Carl Zeiss Microscopy). (Carl Zeiss Microscopy). For imaging a 40x/1,2 W C-Apochromat, Diode lasers with the wavelengths 405 nm, 455 nm, 488 nm, and 561 nm (all 30 mW), GaAsP and Transmitted light (T-PMT) detectors as well as main LSM beamsplitters (MBS 405 and MBS 488/561/633) were used. The experiments were performed at 37 °C and 5% $CO_2$ using an integrated incubation module (Carl Zeiss Microscopy). The Image analysis software ZEN (Carl Zeiss Microscopy) was used. For acquiring confocal images, the following settings were used: Sequential acquisition for the different channels were used. The field of view was set to $512 \times 512$ pixels. For the acquisition the speed was set to max. using line-wise bidirectional scanning, whereby 4x averaging in the Repeat per Line Mode was performed. For acquiring the GFP, mCherry and BFP2 fluorescent signal the following laser setting were used: 6.5 μW for the 488 nm laser (GFP), 79 μW for the 561 nm laser (mCherry) and 13.8 μW for 445 nm laser (BFP2). Brightfield images were acquired using the 561 nm laser with 79 μW and a transmitted light (T-PMT) detector.

## Optogenetic activation in confocal microscopy

ROIs were drawn covering the whole cell area of cells expressing optogenetics tools, as well as bystander controls, using the ROI bleaching function of the microscope software. To control for unwanted activation, not all cells expressing the optogenetics tools within the field of view were activated. For activating the optogenetics tools, the 405 nm laser was used at 456 μW intensity, by setting up a time series acquiring one pre-activation image, before activating selected ROIs with 100 iterations every 7 images or with the indicated activation rounds of the respective experiment. For C11- Bodipy quantification, confocal images were acquired 5- and 65 min upon 3 activation rounds of the optogenetic tools. For the $Ca^{2+}$ experiments, images were acquired every 45 s upon four activation rounds of the optogenetic tools. Experiments were only considered for analysis if all controls worked, using blebbing of cells as proxy for cell death induction throughout the experiments. (1) Illumination of bystander

cells to rule out phototoxicity of the illumination conditions. (2) Non-illumination of transfected cells to check for potential self-activation or construct toxicity. (3) Non-illumination of bystander cells to check for phototoxicity of the live cell imaging time series acquisition (background cell death). If one of these controls failed, meaning that any of the controls cells starts blebbing (indicative for cell death induction) during an experiment, the experiment was excluded from further analysis.

## High throughput optogenetics experiments

Experiments were performed according to the workflow shown in Fig. S1b, unless otherwise indicated. Where indicated, DRAQ7 was added before a pre-activation image was captured for assessing cell death[71]. For activation of the optogenetic tool, an optoPlate-96 was used[32]. For programming the optoPlate-96, optoConfig-96 was used[33]. The following illumination at 465 nm was used: 5 mW/cm² 100%, 4 mW/cm² for 80%, 3 mW/cm² for 60%, 2 mW/cm² for 40% and 1 mW/cm² for 20% LED intensity[32,33]. Kinetics of cell death and C11-Bodipy experiments were assessed using a IncuCyte S3 bioimaging platform (Essen). The pre-activation and post-activation images were acquired with the following settings: Per well, at least three images were captured with 400 ms exposure for the green and red channel. For assessing C11-Bodipy 581/591, an ImageXpress Micro 4 (MD)was used. A 10x Plan Fluor 0.3 NA objective, an Andor Cycla 5.5 (CMOS) camera and the following channels were used: Tl-20 (2 ms), DAPI (50 ms), FITC (200 ms) and TRITC (100 ms) with autocorrection. The experiments were performed at 37 °C and 5% CO₂ using the provided incubation chamber. For the experiment μ-Plate 24 Black ID 14 mm plates (Ibidi) were used.

## Lipid bilayer experiments

All lipids used were purchased from Avanti Polar Lipids (Alabaster, AL). For the bilayer formation liposomes were prepared as small unilamellar vesicles. Lipid compositions were prepared as dry film. The lipids L-α-phosphatidylcholine (Egg, Chicken)(EggPC) and 1,2-dioleoyl-3-trimethylammonium-propane (chloride salt)(DOTAP) were dissolved in chloroform and mixed in a ratio 70:30. The chloroform was evaporated under vacuum for at least 4 h and a maximum of 14 h and additionally exposed to an argon flux for a long term storage at −20 °C. The dry film was hydrated to a final concentration of 10 mg/ml using phosphate-buffered saline (2.7 mM KCl, 1.5 mM KH2PO4, 8 mM Na2HPO4, and 137 mM NaCl, pH 7.2), aliquots of 10 to 40 μl were stored at −20 °C. Aliquots were diluted with 140 μl buffer solution (20 mM HEPES, 140 mM NaCl, pH 7.0) per 10 μl liposome solution volume and sonicated in an ultrasonic bath for 20 min or until clear to form small unilamellar vesicles. Imaging chambers were prepared with plasma cleaned glass slide (length 75 mm, thickness 1.5) and a silicon 8-well chamber (Ibidi). Both slides and chambers were washed prior with 70% ethanol and dried off. A volume of 150 μl (SUV) liposome solution was deposited into one well and filled with buffer to a total volume of 500 μl and CaCl₂ to a total concentration of 3 mM. After an incubation of 10 min at 37 °C floating vesicles and CaCl₂ were washed out by adding and removing 150 μl of buffer solution for at least 10 times. Subsequently 400 μl of the buffer solution was removed and replaced with 300 μl cell suspension containing 2.5 × 10⁴ cells. Then a reverse transfection was performed and on the next day 1 μM C11-Bodipy or 1:500 DRAQ7 was added before the optogenetic activation. Incucyte experiments were performed on a 96 well plate. A volume of 75 μl (SUV) liposome solution was deposited per well and filled with buffer to a total volume of 200 μl and CaCl₂ to a total concentration of 3 mM. After an incubation of 10 min at 37 °C floating vesicles and CaCl₂ were washed out by adding and removing 100 μl of buffer solution for at least 10 times. All steps were identical compared to the confocal setup except that here 150 μl buffer solution was removed and refilled with 100 μl cell suspension containing 1.5 × 10⁴ cells.

## Preparation and oxidation of giant unilamellar vesicles

GUVs were produced by electroformation method[72]. Accordingly, 3 μl of 2.5 mg/mL lipid mixture dissolved in chloroform were spread on platinum wires of the electroformation chamber and allowed to dry, before immersed in 300 μL of 300 mM sucrose. Electroformation was performed using an alternating power generator at 10 Hz, 1.4 V for 2 h, followed by 45 min at 2 Hz, 1.4 V. The lipid mixture is composed of L-α-phosphatidylcholine from egg yolk (egg-PC), 1-palmitoyl-2-arachido-noyl-sn-phosphatidylcholine (PAPC), 1-oleoyl-2-(12 biotinyl(aminodo-decanoyl))-sn-glycero-3-phosphoethanolamine (biotin-PE) in ratio 90:5:5. Oxidation was induced by addition of the photosensitizer 1,9 dimethyl methylene blue (DMMB) 5 mol% (DMMB was dissolved in ethanol (5 mM), Iron (II) percholarte in deionized water (10 mM), and tracked with Bodipy C11 2 mol%, and additionally stained with 1,1'-Dioctadecyl-3,3,3',3'-Tetramethylindodicarbocyanine, 4-Chlorobenzenesulfonate Salt (DiD) 0,05 mol% for differentiation (donor), to a second GUV preparation with the same lipid mixture supplemented with C11-Bodipy 2 mol%, but without the photosensitizer and DiD (acceptor). PBS buffer (2.7 mM KCl, 1.5 mM KH2PO4, 8 mM Na2HPO4, 137 mM NaCl, pH 7.2) was mixed with 0.75 μmol streptavidin for anchoring to biotin-PE and 80 μmol iron (II) perchlorate for a continued lipid radical production. 50 μl of acceptor and donor GUV suspension was added to get a final volume of 250 μl. Confocal fluorescence microscopy was performed using an infinity line scanning microscope (Abberior Instruments) equipped with a UPlanX APO 60x Oil/1.42 NA objective with laser lines to excite at 488, 561 or 640 nm. Donor GUVs were activated by using 20% laser power at 640 nm for a time interval of 1 min (30 x each 2 s) followed by a 20 min image acquisition to acquire oxidation propagation (each 60 s laser power 488 nm 2%, 561 nm 1% and 640 nm 0.5%). Negative controls without presence of iron or in presence of iron, but without photosensitizer activation at 640 nm were additionally performed.

## Image analysis

For assessing the time till blebbing in the confocal experiments the activation time series and post activation time series were imported to Fiji and concentrated into one time series. Afterwards the channels were split to adjust Brightness and Contrast in all channels. After adjusting the channels were merged again to manually assess the time till blebbing of illuminated transfected cells, of illuminated bystander cells, non-illuminated of transfected cells and non-illuminated bystander cells.

To determine the percentage of either transfected, bystander or oxidized C11-Bodipy positive cells of the high-throughput optogenetics experiments, a custom-made software was used[73]. Thereby, the total number of transfected, bystander, as well as cell death-marker positive cells were calculated and normalized taking the absolute number of transfected/bystander cells as 100% divided by the number of double positive cells. For assessing the position and fluorescence level of individual cells, image segmentation was performed using our custom-made software. For assessing the distance between cells, the coordinates of individual cells were used by calculating the individual vector lengths.

## Lipid peroxidation experiments

Cells were transfected with Opto-GPX4Deg and incubated for 16 h. On the next day the cells were stained with 1 μM C11-Bodipy for one hour and treated with 10 μM zVAD. Then Opto-GPX4Deg expressing cells and bystander controls were activated for 3 times (see Optogenetic activation in confocal microscopy) and the fluorescence level of C11-Bodipy for individual cells (red and green) were assessed 5- and 65-min post-activation and corrected for the cell size and local background subtraction. The oxidation ratio of C11-Bodipy was calculated as an indicator of lipid peroxidation according to the following formula:

oxidation ratio = red + green fluorescence/green fluorescence. Thereby the red fluorescence corresponds to the reduced fraction of the probe and was estimated based on the fluorescence intensity per pixel from the red image and the green fluorescence corresponds to the oxidized fraction and was calculated based on the fluorescence intensity per pixel from the green channel fluorescence images. Those values were estimated based on the fluorescence intensity per pixel from the red and green channels of the fluorescence images using Fiji or our custom-made software.

## Ca²⁺ wave measurements

For assessing $Ca^{2+}$ fluxes in the confocal experiments, the activation time series and post activation time series were imported to Fiji and concentrated into one time series. Afterwards, ROIs of cells undergoing ferroptosis (activated, Opto-GPX4Deg expressing cells), as well as for their neighboring cells in the first to the third rows were drawn. Then the channels were split and the green fluorescence signal was measured for every time point using Fijis multi measure function.

## Immunoflouresence staining

$3 \times 10^4$ cells were seeded on a 12 mm cover slip and on the next day either transfected with a respective siRNA or expression plasmid. The cells were incubated for 48 h before fixation in 4% formaldehyde for 15 min on 25 °C. The specimen were rinsed three times in PBS before blocking them in 1X PBS/BSA/0.3% Triton™ X-100 buffer for 60 min. Subsequently the blocking buffer was aspirated and the specimen were incubated over night at 4 °C in 1:1000 Anti-YAP D8H1X (Cell Signaling) diluted in 1X PBS/1% BSA/0.3% Triton™ X-100 buffer (Antibody Dilution Buffer). Then the specimen were rinsed 3 times in PBS for 5 min before incubating them 1:1000 Goat anti-Rabbit IgG (H + L) Cross-Adsorbed Secondary Antibody, Alexa Fluor™ 488 (Thermo Fisher Scientific, A-11008, polyclonal, Lot-2420730). Antibody Dilution Buffer for 1 h protected from light. Afterwards the specimen were washed 3 times in PBS and Hoechst33342 (Thermo Fisher) counterstaining was performed for 5 min (1 μg/ml). Afterwards the specimen were 3 times in PBS after which the samples were mounted for imaging.

## Immunoblotting

Cells were lysed in RIPA buffer supplemented with protease inhibitor (Roche). To load equal amounts of proteins on a 12% SDS-PAGE, a Bradford assay was performed. The transfer onto a nitrocellulose membrane (Merck Millipore) was performed using a the Turboblot (BioRad) or a wet transfer system (BioRad). The membrane was blocked with 5% milk in PBS-T (0.1% Tween) for 1 h at room temperature and incubated over night with the respective primary antibodies: 1:1000 ß-tubulin (Santa Cruz Biotechnology, # sc-55529, clone G8, Lot-F2812), 1:1000 Anti-Glutathione Peroxidase 4 (# ab125066, clone EPNCIR144, Lot-GR3369574-7) (Abcam), 1:1000 Anti-GFP antibody (# ab290, polyclonal) (Abcam) and 1:1000 α-catenin (Sigma-Aldrich, # C2081, polyclonal, Lot-0000145845), 1:1000 α-E-Catenin (D9R5E, #36611, Lot-2) (Cell Signaling), 1:1000 Anti-FSP1 (PTG Lab, # 20886-1-AP, polyclonal, Lot-00101489), 1:1000 Anti-YAP (Cell Signaling, #14074, clone D8H1X, Lot-5), 1:1000 Anti-phospho-YAP (Ser127) (Cell Signaling, #13008, clone D9W2I, Lot-6), 1:200 Anti-GAPDH (Santa Cruz Biotechnology, sc-47724, clone 0411, Lot-12420), 1:1000 Anti-VDAC2 (PTG Lab, 11663-1-AP, polyclonal), 1:1000 Anti-ß-Actin (Santa Cruz Biotechnology, sc-47778, clone C4, Lot-E0720), Anti-mCherry E5D8F (Cell Signaling). The membrane was then washed five times for 5 min in PBS-T at room temperature and incubated with the secondary antibody at room temperature for one hour in the dark. The following secondary antibodies were used: 1:5000 IRDye 800CW Donkey anti-Mouse IgG Secondary Antibody (LI-COR, # 926-32212), 1:10000 IRDye 680RD Donkey-anti-Rabbit Antibody (LI-COR, # 926-68023). The membrane was washed five times for 5 min with PBS-T and the fluorescence signals were captured by an Odyssey DLx (LI-COR). Western Blots were quantified using the Image Studio Lite Software.

## Optogenetic activation for lipidomics

On day 0, $1 \times 10^5$ HeLa cells/well were seeded into 6-well plates. On day 2, HeLa cells were transfected (see above) either with Opto-Ctrl, Opto-GPX4Deg or not transfected. Bulks were illuminated or not for 60 min with 100% 465 nm LED intensity using an optoPlate-96. After 48 h, the cells were harvested by trypsinization and samples were prepared for lipidomics (see below).

## Quantification of oxidized glycerophospholipids

Levels of oxidized phosphatidylcholine (PC) and phosphatidylethanolamine (PE) species were determined by Liquid Chromatography coupled to Electrospray Ionization Tandem Mass Spectrometry (LC-ESI-MS/MS)[4]. 1–2 million cells were resuspended in 300 μl of an ice-cold solution of 100 μM diethylenetriaminepentaacetic acid (DTPA) in PBS. An aliquot of the cell suspension was used for the determination of the protein content using bicinchoninic acid. To 100 μl of the cell suspension, 2.4 ml of an ice-cold solution of 1.5 mg/ml triphenylphosphine and 0.005% butylated hydroxytoluene in methanol were added. The mixture was incubated for 20 min in a shaking bath at room temperature. Afterwards, 1 ml of the 100 μM DTPA in PBS solution, 1.25 ml of chloroform and internal standards (10 pmol 1,2-dimyristoyl-sn-glycero-3-phosphocholine (DMPC) and 10 pmol 1,2-dimyristoyl-sn-glycero-3-phosphoethanolamine (DMPE)) were added. The samples were vortexed for 1 min and incubated at −20 °C for 15 min. After adding 1.25 ml of chloroform and 1.25 ml of water, the mixture was vortexed vigorously for 30 s and then centrifuged ($4000 \times g$, 5 min, 4 °C) to separate layers. The lower (organic) phase was transferred to a new tube and dried under a stream of nitrogen. The residues were resolved in 150 μl of methanol and transferred to autoinjector vials.

LC-MS/MS analysis was performed using a Shimazdu Nexera UHPLC system[4,74] by injecting 20 μl of sample onto a Core-Shell Kinetex C18 column (150 mm × 2.1 mm ID, 2.6 μm particle size, 100 Å pore size, Phenomenex) at 30 °C and with detection using a QTRAP 6500 triple quadrupole/linear ion trap mass spectrometer (SCIEX). The LC (Nexera X2 UHPLC System, Shimadzu) was operated at a flow rate of 0.2 ml/min with a mobile phase of 1 mM ammonium acetate in methanol/acetonitrile/water 60:20:20 (v/v/v) (solvent A) and 1 mM ammonium acetate in methanol (solvent B). Oxidized PC and PE species were eluted with the following gradient: initial, 50% B; 10 min, 100% B; 40 min, 100% B; 40.1 min, 50% B; 45 min, 50% B. Lipid species were monitored in the negative ion mode with their specific Multiple Reaction Monitoring (MRM) transitions (Table 1)[73]. The instrument settings for nebulizer gas (Gas 1), turbogas (Gas 2), curtain gas, and collision gas were 40 psi, 30 psi, 35 psi, and medium, respectively. The interface heater was on, the Turbo V ESI source temperature was 500 °C, and the ionspray voltage was −4.5 kV. The values for declustering potential (DP), entrance potential (EP), collision energy (CE), and cell exit potential (CXP) of the different MRM transitions are listed in Table 1.

The LC chromatogram peaks of oxidized PC and PE species and the internal standards were integrated using the MultiQuant 3.0.2 software (SCIEX). Oxidized PC and PE species were quantified by normalizing their peak areas to those of the internal standards. The normalized peak areas were then normalized to the protein content of the cell suspension. Fold change in the heat map calculated by dividing Area ratios/μg protein of different oxidized lipid species upon illumination divided by the non-illuminated control for the individual replicates and the average.

## Quantification of fatty acids

To 50 μl of the cell suspension mentioned above, 50 μl of water, 500 μl of methanol, 250 μl of chloroform, and internal standard (1 μg palmitic-d31 acid) were added. The mixture was sonicated for 5 min, and lipids were extracted in a shaking bath at 48 °C for 1 h. Glycerolipids were degraded by alkaline hydrolysis adding 75 μl of 1 M potassium

**Table 1 | MRM transitions and compound-specific parameters of oxidized PC and PE species**

| Q1 Mass (Da) | Q3 Mass (Da) | Time (msec) | ID | DP (volts) | EP (volts) | CE (volts) | CXP (volts) |
|---|---|---|---|---|---|---|---|
| 634.5 | 227.2 | 50 | PE(14:0_14:0) | −50 | −10 | −38 | −11 |
| 662.5 | 227.2 | 50 | PC(14:0_14:0) | −50 | −10 | −38 | −11 |
| 810.6 | 347.2 | 50 | PC(16:0a_22:4(O)) | −50 | −10 | −38 | −11 |
| 764.6 | 317.2 | 50 | PC(16:0p_20:5(O)) | −50 | −10 | −38 | −11 |
| 808.7 | 361.2 | 50 | PC(16:0p_22:5(2 O)) | −50 | −10 | −38 | −11 |
| 790.6 | 343.2 | 50 | PC(16:0p_HDoHE) | −50 | −10 | −38 | −11 |
| 766.6 | 319.2 | 50 | PC(16:0p_HETE) | −50 | −10 | −38 | −11 |
| 782.6 | 347.2 | 50 | PE(16:0a_22:4(O)) | −50 | −10 | −38 | −11 |
| 804.7 | 343.2 | 50 | PE(16:0a_HDoHE) | −50 | −10 | −38 | −11 |
| 736.6 | 317.2 | 50 | PE(16:0p_20:5(O)) | −50 | −10 | −38 | −11 |
| 766.6 | 347.2 | 50 | PE(16:0p_22:4(O)) | −50 | −10 | −38 | −11 |
| 780.6 | 361.2 | 50 | PE(16:0p_22:5(2 O)) | −50 | −10 | −38 | −11 |
| 762.6 | 343.2 | 50 | PE(16:0p_HDoHE) | −50 | −10 | −38 | −11 |
| 738.6 | 319.2 | 50 | PE(16:0p_HETE) | −50 | −10 | −38 | −11 |
| 782.6 | 319.2 | 50 | PE(18:0a_HETE) | −50 | −10 | −38 | −11 |
| 808.7 | 347.2 | 50 | PE(18:1a_22:4(O)) | −50 | −10 | −38 | −11 |
| 778.6 | 343.2 | 50 | PE(18:1a_HDoHE) | −50 | −10 | −38 | −11 |
| 792.6 | 347.2 | 50 | PE(18:1p_22:4(O)) | −50 | −10 | −38 | −11 |
| 788.6 | 343.2 | 50 | PE(18:1p_HDoHE) | −50 | −10 | −38 | −11 |
| 764.6 | 319.2 | 50 | PE(18:1p_HETE) | −50 | −10 | −38 | −11 |

**Table 2 | MRM transitions and compound-specific parameters of fatty acids**

| Q1 Mass (Da) | Q3 Mass (Da) | Time (msec) | ID | DP (volts) | EP (volts) | CE (volts) | CXP (volts) | Quantifier |
|---|---|---|---|---|---|---|---|---|
| 227.0 | 227.0 | 5 | FA 14:0 | −75 | −10 | −25 | −13 | X |
| 227.0 | 227.1 | 5 | FA 14:0 | −75 | −10 | −30 | −13 | |
| 225.0 | 225.0 | 5 | FA 14:1 | −75 | −10 | −8 | −13 | X |
| 225.0 | 225.1 | 5 | FA 14:1 | −75 | −10 | −23 | −13 | |
| 255.0 | 255.0 | 5 | FA 16:0 | −80 | −10 | −35 | −13 | X |
| 255.0 | 255.1 | 5 | FA 16:0 | −80 | −10 | −37 | −13 | |
| 253.0 | 253.0 | 5 | FA 16:1 | −78 | −10 | −30 | −13 | X |
| 253.0 | 253.1 | 5 | FA 16:1 | −78 | −10 | −34 | −13 | |
| 283.1 | 283.1 | 5 | FA 18:0 | −87 | −10 | −35 | −7 | X |
| 283.1 | 283.2 | 5 | FA 18:0 | −87 | −10 | −37 | −7 | |
| 281.1 | 281.1 | 5 | FA 18:1 | −80 | −10 | −37 | −7 | X |
| 281.1 | 281.0 | 5 | FA 18:1 | −80 | −10 | −39 | −7 | |
| 279.0 | 279.1 | 5 | FA 18:2 | −80 | −10 | −32 | −7 | X |
| 279.0 | 279.0 | 5 | FA 18:2 | −80 | −10 | −34 | −7 | |
| 277.1 | 277.1 | 5 | FA 18:3 | −75 | −10 | −10 | −7 | X |
| 277.1 | 277.0 | 5 | FA 18:3 | −75 | −10 | −23 | −7 | |
| 309.1 | 309.1 | 5 | FA 20:1 | −90 | −10 | −8 | −9 | X |
| 309.1 | 309.0 | 5 | FA 20:1 | −90 | −10 | −30 | −9 | |
| 307.1 | 307.2 | 5 | FA 20:2 | −85 | −10 | −8 | −9 | X |
| 307.1 | 307.1 | 5 | FA 20:2 | −85 | −10 | −29 | −9 | |
| 305.1 | 305.1 | 5 | FA 20:3 | −80 | −10 | −8 | −9 | X |
| 305.1 | 305.0 | 5 | FA 20:3 | −80 | −10 | −26 | −9 | |
| 303.1 | 303.1 | 5 | FA 20:4 | −70 | −10 | −10 | −9 | X |
| 303.1 | 303.0 | 5 | FA 20:4 | −70 | −10 | −21 | −9 | |
| 337.1 | 337.1 | 5 | FA 22:1 | −80 | −10 | −8 | −9 | X |
| 337.1 | 337.0 | 5 | FA 22:1 | −80 | −10 | −30 | −9 | |
| 335.1 | 335.1 | 5 | FA 22:2 | −80 | −10 | −8 | −11 | X |
| 335.1 | 335.0 | 5 | FA 22:2 | −80 | −10 | −30 | −11 | |
| 327.1 | 327.0 | 5 | FA 22:6 | −40 | −10 | −6 | −13 | X |
| 327.1 | 327.1 | 5 | FA 22:6 | −40 | −10 | −15 | −13 | |
| 365.1 | 365.1 | 5 | FA 24:1 | −97 | −10 | −8 | −11 | X |
| 365.1 | 365.0 | 5 | FA 24:1 | −97 | −10 | −33 | −11 | |
| 286.1 | 286.1 | 5 | d31_FA 16:0 | −80 | −10 | −10 | −7 | X |
| 286.1 | 286.2 | 5 | d31_FA 16:0 | −80 | −10 | −27 | −7 | |

hydroxide in methanol. After 5 min of sonication, the extract was incubated for 1.5 h at 37 °C, and then neutralized with 6 µl of glacial acetic acid. 2 ml of chloroform and 4 ml of water were added to the extract which was vortexed vigorously for 30 s and then centrifuged (4000 × g, 5 min, 4 °C) to separate layers. The lower (organic) phase was transferred to a new tube, and the upper phase extracted with additional 2 ml of chloroform. The combined organic phases were dried under a stream of nitrogen. The residues were resolved in 300 µl of acetonitrile/water 2:1 (v/v) and sonicated for 5 min. After centrifugation (12,000 × g, 2 min, 4 °C), 40 µl of the clear supernatants were transferred to autoinjector vials.

Fatty acid levels were determined by LC-ESI-MS/MS[75]. 10 µl of sample were loaded onto a Core-Shell Kinetex Biphenyl column (100 mm × 3.0 mm ID, 2.6 µm particle size, 100 Å pore size, Phenomenex), and fatty acids were detected using a QTRAP 6500 triple quadrupole/linear ion trap mass spectrometer (SCIEX). The LC (Nexera X2 UHPLC System, Shimadzu) was operated at 40 °C and at a flow rate of 0.7 ml/min with a mobile phase of 5 mM ammonium acetate and 0.012 % acetic acid in water (solvent A) and acetonitrile/isopropanol 80:20 (v/v) (solvent B). Fatty acids were eluted with the following gradient: initial, 55% B; 4 min, 95% B; 7 min, 95% B; 7.1 min, 55% B; 10 min, 55% B.

Fatty acids were monitored in the negative ion mode using "pseudo" Multiple Reaction Monitoring (MRM) transitions (Table 2)[75]. The instrument settings for nebulizer gas (Gas 1), turbogas (Gas 2), curtain gas, and collision gas were 60 psi, 90 psi, 40 psi, and medium, respectively. The interface heater was on, the Turbo V ESI source temperature was 650 °C, and the ionspray voltage was −4 kV. The values for declustering potential (DP), entrance potential (EP), collision energy (CE), and cell exit potential (CXP) of the different MRM transitions are listed in Table 2.

The LC chromatogram quantifier peaks of the endogenous fatty acids and the internal standard palmitic-d31 acid were integrated using the MultiQuant 3.0.2 software (SCIEX). Endogenous fatty acids were quantified by normalizing their peak areas to those of the internal standard. Fold change in the heat map calculated by dividing Area ratios of different fatty acids upon illumination divided by the non-illuminated control for the individual replicates and the average.

### MEF co-culture with ferroptotic supernatants

Cell culture supernatants coming from ferroptotic or control cells were first produced in two independent genetic systems: (i) mouse embryonic fibroblasts (MEFs) with inducible GPX4 knockout[39] were treated with 1 µM 4-hydroxytamoxifen or DMSO for 72 h; (ii) murine small cell lung cancer (SCLC) GPX4 knockout or control cells[74] were placed upon ferrostatin withdrawal for 16 h. Supernatants were collected, spun down at 1200 rpm for 5 min and transferred onto wildtype (WT) MEFs seeded one day in advance with or without 0.1 µM RSL3 or 1 µM Erastin +/-1 µM ferrostatin.

### Statistical analysis

For assessing significant differences different statistical tests were performed depending on the number of compared groups as well as on the factors considered in the experiment. For comparing two groups or conditions non-parametric t-tests were performed. When more than one t-test was performed within on graph, they were corrected for multiple comparisons using a multiple t-test. When more than two groups or conditions were analyzed, a one-way ANOVA was performed, when one factor was compared, whereas and a three-way ANOVA was performed, when three factors were compared. For correcting for multiple comparisons, the Tukey's multiple comparison test was performed. For performing the statistical tests, Grappad Prism 7.03 was used. For assessing the clustering tendency of the data set, Hopkins statistics was performed using freely available Code on Github[76].

### Reporting summary

Further information on research design is available in the Nature Portfolio Reporting Summary linked to this article.

## Data availability

Source data are provided with this paper. Lipidomics data have been deposited in the Zenodo data repository and can be downloaded via https://doi.org/10.5281/zenodo.14905646. All other data supporting the results reported here are available upon request. Access to the data will be granted upon request for academic or non-commercial research purposes, provided that the requester agrees to maintain confidentiality (if applicable), properly cite the source, comply with any legal or ethical restrictions, and that fulfilling the request does not impose an undue burden on the authors. Source data are provided with this paper.

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

## Acknowledgements
We thank J. Riemer (Univ. Cologne), Uris Ros and Lohans Pedrera for helpful discussions. This project was funded by the Deutsche Forschungsgesellschaft (DFG, German Research foundation), SFB1403 – project no. 414786233, and partially SPP2306 – project no. GA1641/7-1 and by the Bundesministerium für Bildung und Forschung (BMBF) project 16LW0213. We also thank the CECAD Imaging and Lipidomics/Metabolomics facilities as well as their staff members for their support. F.I.Y. and S.v.K. were funded through funding through SPP2306, project ID 461704389; CRC1403, project ID 414786233; CRC1310, project ID 325931972; CRC1399, project ID 413326622 and CRC1530, project ID 455784452, by the German Research Foundation (Deutsche Forschungsgesellschaft, DFG), as well as by the BMBF (InCa-01ZX2201A) and CANTAR, program " Netzwerke 2021" of the Ministry of Culture and Science of the State of Northrhine Westphalia, Germany and a project grant (A07) funded by the center for molecular medicine cologne (CMMC).

## Author contributions
B.F.R. performed experiments and analyzed data. M.R.H.V. built custom software and analyzed experiments. S.L.N. and B.F.R. designed and performed the lipid bilayer experiment. S.L.N. performed the experiments with GUVs. L.S. generated the HeLa α-Catenin KO cells. F.I.Y. and S.v.K. designed and conducted the supernatant experiment. M.R. and C.M.N. designed the cell-cell contact experiments regarding α-catenin and blebbistatin, as well as the E-cadherin, N-cadherin, and P-cadherin overexpression experiments. They also contributed to the interpretation of the acquired data. All authors contributed to experimental design and manuscript writing. A.J.G.-S. conceived the project, supervised research and wrote the manuscript together with B.F.R.

## Funding

## Competing interests
The authors declare no competing interests.
