## [Transparent Peer Review file · Nature Communications]

Ferroptosis propagates to neighboring cells via plasma membrane contacts

Corresponding Author: Professor Ana J García-Sáez

Version 0:

Reviewer comments:

Reviewer #1

(Remarks to the Author)

This manuscript, now at Nature Communications, includes important controls, such as the important Figure S9. While the effect protective effect of the incomplete alpha-catenin knockout, when stimulated with RSL3, is not as strong as the the effect observed with the Opto-GPX4Deg system, it yet is clearly significant.

A second important figure is S12 that clearly helped to convincingly demonstrate the propagation between the GUVs. While this does not necessarily reflect cellular systems, it still represents an important finding.

With these two major concerns worked out in detail, I recommend publication of this study.

Reviewer #2

(Remarks to the Author)

The authors have sufficiently addressed my concerns.

I would still suggest to include this recent Nature paper, which shows contact independent ferroptosis spreading, in the discussion:

Co, H. K. C., Wu, C.-C., Lee, Y.-C. & Chen, S. Emergence of large-scale cell death through ferroptotic trigger waves. Nature 631, 654–662 (2024).

REVIEWERS' COMMENTS

Reviewer #1 (Remarks to the Author):

This manuscript, now at Nature Communications, includes important controls, such as the important Figure S9. While the effect protective effect of the incomplete alpha-catenin knockout, when stimulated with RSL3, is not as strong as the the effect observed with the Opto-GPX4Deg system, it yet is clearly significant.

A second important figure is S12 that clearly helped to convincingly demonstrate the propagation between the GUVs. While this does not necessarily reflect cellular systems, it still represents an important finding.

With these two major concerns worked out in detail, I recommend publication of this study.

We thank the reviewer for recommending our manuscript for publication.

Reviewer #2 (Remarks to the Author):

The authors have sufficiently addressed my concerns.

I would still suggest to include this recent Nature paper, which shows contact independent ferroptosis spreading, in the discussion:

Co, H. K. C., Wu, C.-C., Lee, Y.-C. & Chen, S. Emergence of large-scale cell death through ferroptotic trigger waves. Nature 631, 654–662 (2024).

We followed the reviewers suggestion and added the respective study in our discussion.